# CD73-mediated adenosine production by CD8 T cell-derived extracellular vesicles constitutes an intrinsic mechanism of immune suppression

Enja Schneider [1,15], Riekje Winzer [1,15✉], Anne Rissiek[1], Isabell Ricklefs[2,3], Catherine Meyer-Schwesinger[4], Franz L. Ricklefs [5], Andreas Bauche[6], Jochen Behrends [7], Rudolph Reimer[8], Santra Brenna [9], Hauke Wasielewski [1], Melchior Lauten [10], Björn Rissiek [9], Berta Puig [9], Filippo Cortesi [11,12], Tim Magnus [9], Ralf Fliegert[6], Christa E. Müller [13], Nicola Gagliani [11,12,14] & Eva Tolosa [1✉]

Immune cells at sites of inflammation are continuously activated by local antigens and cytokines, and regulatory mechanisms must be enacted to control inflammation. The stepwise hydrolysis of extracellular ATP by ectonucleotidases CD39 and CD73 generates adenosine, a potent immune suppressor. Here we report that human effector CD8 T cells contribute to adenosine production by releasing CD73-containing extracellular vesicles upon activation. These extracellular vesicles have AMPase activity, and the resulting adenosine mediates immune suppression independently of regulatory T cells. In addition, we show that extracellular vesicles isolated from the synovial fluid of patients with juvenile idiopathic arthritis contribute to T cell suppression in a CD73-dependent manner. Our results suggest that the generation of adenosine upon T cell activation is an intrinsic mechanism of human effector T cells that complements regulatory T cell-mediated suppression in the inflamed tissue. Finally, our data underscore the role of immune cell-derived extracellular vesicles in the control of immune responses.

[1] Department of Immunology, University Medical Center Hamburg-Eppendorf, 20246 Hamburg, Germany. [2] Division of Pediatric Pneumology & Allergology, University Medical Center Schleswig-Holstein, 23538 Lübeck, Germany. [3] Airway Research Center North, Member of the German Center for Lung Research, Lübeck, Germany. [4] Department of Cellular and Integrative Physiology, University Medical Center Hamburg-Eppendorf, 20246 Hamburg, Germany. [5] Department of Neurosurgery, University Medical Center Hamburg-Eppendorf, 20246 Hamburg, Germany. [6] Department of Biochemistry and Molecular Cell Biology, University Medical Center Hamburg-Eppendorf, 20246 Hamburg, Germany. [7] Core Facility Fluorescence Cytometry, Research Center Borstel, 23845 Borstel, Germany. [8] Technology Platform Microscopy and Image Analysis, Heinrich Pette Institute/Leibniz Institute for Experimental Virology, 20251 Hamburg, Germany. [9] Department of Neurology, University Medical Center Hamburg-Eppendorf, 20246 Hamburg, Germany. [10] Department of Pediatrics and Adolescent Medicine, University of Lübeck, 23538 Lübeck, Germany. [11] I. Department of Medicine, University Medical Center Hamburg-Eppendorf, 20246 Hamburg, Germany. [12] Department of General, Visceral and Thoracic Surgery, University Medical Center Hamburg-Eppendorf, 20246 Hamburg, Germany. [13] Department of Pharmaceutical & Medicinal Chemistry, University of Bonn, 53121 Bonn, Germany. [14] Immunology and Allergy Unit, Department of Medicine, Solna, Karolinska Institute and University Hospital, Stockholm, Sweden. [15] These authors contributed equally: Enja Schneider, Riekje Winzer. ✉email: r.winzer@uke.de; etolosa@uke.de

Immune cell activation, cellular stress, or metabolic changes during inflammation favor the release of ATP into the extracellular space. High extracellular ATP is a danger signal for immune cells and is swiftly metabolized by ATP-degrading enzymes. Among them, the ectonucleotidase CD39 dephosphorylates ATP and ADP to AMP, which is subsequently converted to adenosine by Ecto-5′-nucleotidase (CD73)[1]. The amount of available extracellular adenosine is further determined by the rate of adenosine deaminase (ADA)-mediated degradation, and by cellular uptake through nucleoside transporters. Activation of the adenosine receptor subtype $A_{2A}$, the predominantly expressed adenosine receptor in T cells (ImmGen database consortium[2]), results in a rise of intracellular cAMP, leading to decreased T cell activation and effector function[3,4].

Increased adenosine signaling limits mucosal inflammation[5] and improves disease in several animal models of autoimmunity[6,7]. Deletion of the $A_{2A}$ receptor enhances gastritis in *Helicobacter*-infected mice[8] and exacerbates inflammation in the early stages of experimental autoimmune encephalomyelitis[9]. In humans, high ADA activity has been documented in the serum of patients with autoimmune diseases[7,10]. Consequently, the adenosine-generating enzyme CD73 plays a protective role in the animal models of arthritis[11] and colitis[12].

The control of immune responses is crucial to prevent inflammation-induced damage to healthy tissue. FOXP3[+] regulatory T cells (Tregs) are essential to maintain peripheral tolerance to self-antigens and play a pivotal role in terminating an immune response by inhibiting T cell proliferation and effector function. Tregs use an array of suppressive mechanisms to restore immune homeostasis, including the production of anti-inflammatory cytokines, engagement of co-inhibitory receptors, and the modulation of effector T cell metabolism[13]. Murine Tregs express CD39 and high levels of CD73 on the cell surface to degrade ATP and produce adenosine, which in turn has a dual effect; inhibiting effector T cells[14] and enhancing the suppressive capacity of Tregs[15]. In the human T cell compartment, however, CD73 is expressed on the surface of most naïve CD8 T cells and in a small proportion of mature CD4 and CD8 memory T cells, but it is almost absent on Tregs[16–18]. Therefore, co-expression of CD73 with CD39 on Tregs is a rare event[17,19], challenging the concept of adenosine generation as a suppression mechanism used by Tregs in humans. While several studies demonstrate the importance of CD39 expression on human Tregs for their suppressive capacity[17,20,21], the evidence for an essential role of CD73 on Tregs is controversial[22–24].

Considering the low expression of CD73 on human Tregs, the question arises of how immunosuppressive adenosine is generated in the human system under conditions of inflammation. Our data reveal that CD73 contained in extracellular vesicles (EVs) derived from activated CD8 T cells is sufficient to degrade AMP and dampen T cell proliferation and function. This T cell-intrinsic mechanism, in concerted action with a high ATPase activity of Tregs, mediates the production of adenosine from ATP, and warrants sufficient immune suppression. Moreover, we find that EVs isolated from the synovial fluid (SF) of patients with juvenile idiopathic arthritis (JIA) induce T cell suppression in a CD73-dependent manner, underscoring the relevance of CD73 on EVs in the control of inflammation.

## Results

### Human regulatory T cells do not generate sufficient adenosine to suppress T cell proliferation and function

The concentration of pericellular ATP increases upon T cell activation[25] and ATP is quickly metabolized to adenosine by the ectonucleotidases CD39 and CD73. Activation of $A_2$ receptors on immune cells increases intracellular cAMP levels, resulting in decreased T cell activation, and effector function[4] (Fig. 1a). To assess the effect of adenosine receptor activation in human primary T cells, we first added EHNA, an inhibitor of ADA that prevents the degradation of adenosine. In the presence of EHNA, we observed a 20% reduction in T cell proliferation compared to the untreated control (Fig. 1b). Similarly, the metabolically stable, nonselective adenosine analogs 5′-N-ethylcarboxamidoadenosine (NECA) and 2-chloroadenosine (CADO) led to a concentration-dependent decrease in activation (measured as a percentage of CD25[+] cells out of CD4 T cells) and proliferation (Fig. 1c, d), demonstrating the suppressive effect of adenosine receptor activation on human T cells.

Adenosine is generated from ATP by the concerted action of CD39 and CD73, and co-expression of these ectonucleotidases is a hallmark of murine Foxp3[+] Tregs and allows them to generate immunosuppressive adenosine[14]. Even though this pathway is widely accepted to be valid also in the human system, gene expression data from the Human Protein Atlas[26,27] and the ImmGen project[2,28] show that the expression of *NT5E*, the gene encoding CD73, is divergent in different immune cell compartments of the two species. According to gene expression data, all murine T cell subpopulations express *Nt5e*, while in the human peripheral T cell compartment CD8 T cells have a high *NT5E* expression and the expression in Tregs is much lower. Our data confirm that nearly all peripheral Tregs in mice express high levels of CD73 on the cell surface, and two-thirds of the cells co-express CD39 (Supplementary Fig. 1), fulfilling the enzymatic requirements for adenosine generation. To systematically explore CD39 and CD73 expression on human peripheral T cells, we measured cell surface expression of these two ectonucleotidases by flow cytometry (Fig. 1e, f). We found that only a minimal frequency of human peripheral Tregs express CD73 (average of 3%, ranging from 0.8 to 6%, Fig. 1f). CD39 is expressed on 10 to 70% of Tregs, depending on the genotype of the donor[17,29], and on around 5% of nonactivated conventional CD4 (CD4con, defined as non-Treg CD4 T cells) and CD8 T cells. CD73, in contrast, is expressed on ~20 to 60% of the CD8 T cells, and on less than 20% of CD4con T cells. Given the low frequency of Tregs expressing CD73, co-expression of both ectoenzymes is a rare event even in donors with high CD39 expression (Fig. 1e, f), questioning the relevance of Treg-derived adenosine for immune suppression in the human system. To address this point experimentally, we performed an in vitro suppression assay using different ratios of CD4con T cells:Treg (Fig. 1g), considering that the physiological proportion of Tregs is around 10% of CD4 T cells[30]. As responder cells, we used sorted CD73[−] CD4con T cells to prevent the production of adenosine by CD73[+] cells other than Tregs (Supplementary Fig. 2). In these specific conditions, we did not observe an effect of Tregs on suppression at any ratio. When we added ATP to mimic the inflammatory milieu, we observed maximal suppression of T cell activation, proliferation, and IFNγ production at a high Treg ratio (1:0.5 CD4con T cells to Tregs, fivefold the physiological concentration). The addition of recombinant CD73 had no further suppressive effect, indicating that there was enough CD73 in the system (in this donor in particular 2% of Tregs were CD73[+]) to produce adenosine. At a low Treg ratio (1:0.125 CD4con T cells to Tregs, similar to physiological conditions), Tregs could only induce partial suppression in the presence of ATP. In combination with recombinant CD73, though, expression of CD25 was reduced to the minimum, and proliferation and IFNγ production were completely abolished (Fig. 1g). These data indicate that at a physiological CD4con:Treg ratio, there is not enough Treg-derived AMPase activity to generate adenosine that mediates significant suppression.

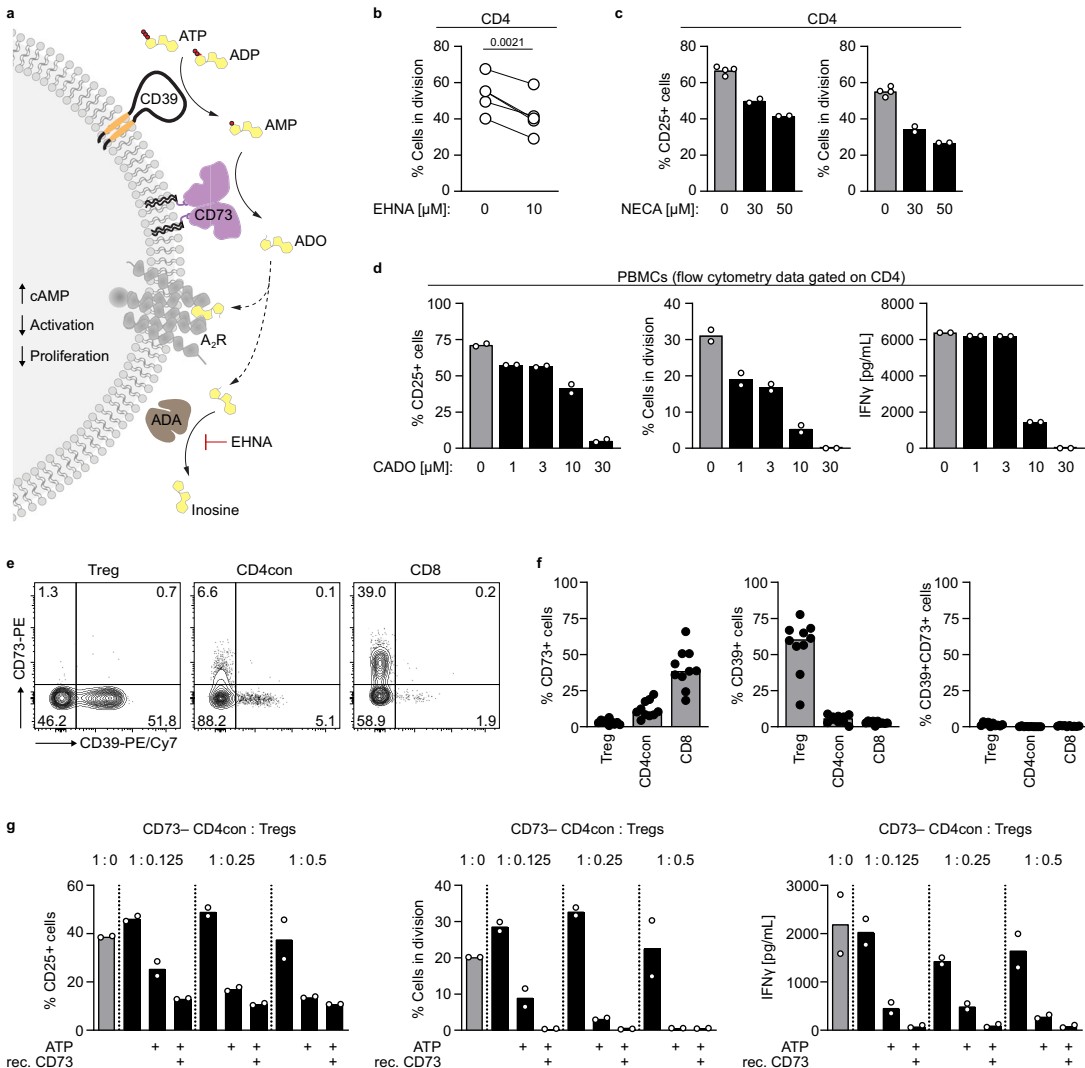

**Fig. 1 Adenosine generated in the course of T cell activation prevents an exacerbated T cell response. a** Schematic representation for the degradation of ATP to adenosine by the ectonucleotidases CD39 and CD73. **b–d** PBMCs or CD4 T cells were stimulated with αCD3/αCD28 and treated with **b** the ADA inhibitor EHNA or different concentrations of adenosine receptor agonists **c** NECA, and **d** CADO. CD25 expression and proliferation were measured after 3 to 4 days by flow cytometry. Data were shown for **b** five donors or **c**, **d** one representative donor (mean of technical replicates). **e**, **f** CD73 and CD39 expression on human T cell subsets in **e** representative dot plots and **f** as a summary for ten donors (median). **g** CD73⁻ CD4con T cells were stimulated with αCD3/αCD28 in the presence of the ADA inhibitor EHNA (10 μM) and incubated with Tregs, ATP (50 μM) and recombinant CD73 (15 ng/mL) as indicated. The ratios of CD4con T cells to Tregs were 1:0.125 to 1:0.5. CD25 expression and proliferation were measured after 4 days by flow cytometry. IFNγ production was determined by ELISA in the cell culture supernatant on day 4. Data were shown for one representative donor out of four analyzed (mean of technical duplicates). A two-tailed paired *t*-test was used to compare untreated and treated samples in **b**.

**CD73-mediated AMPase activity by Tregs is dispensable for the control of CD4 T cell proliferation and function**. In humans, CD73 expression is less frequent in Tregs than in CD4con and CD8 T cells (Fig. 1f). We hypothesized that AMPase activity derived from nonregulatory T cells contributes to adenosine production and immune suppression. To test this, we stimulated CD4con T cells and added Tregs at different ratios. As expected, the addition of Tregs at a very high CD4con:Treg ratio (1:2) resulted in a decrease of T cell activation, and proliferation by 30 to 50%, respectively. We observed a dramatic reduction when exogenous AMP was added to the cell culture to ensure an equal amount of substrate for CD73 in all conditions. Importantly, this effect was independent of the Treg-derived AMPase activity (Fig. 2a). We reasoned that the likely source of AMPase activity in our system could be CD73 from the responder T cells themselves. To test this, we sorted CD4con T cells into CD73⁻

and CD73⁺ and stimulated them in the presence of AMP, but without Tregs (Fig. 2b). As predicted, CD73⁺-sorted cells were less activated and proliferated at lower levels after incubation with AMP, and this effect was reversed by adding the specific CD73 inhibitor PSB-14685. Importantly, the addition of AMP to CD73⁻-sorted CD4con T cells did not have any suppressive effect on activation or proliferation (Fig. 2b), indicating that CD73 on the responder T cells is necessary for the production of adenosine. Addition of recombinant CD73 to AMP-treated CD73⁻ CD4con T cells reestablished the suppressive effect, demonstrating that exogenous CD73 functionally compensates the lack of CD73 in the responder cells (Fig. 2c). Of note, recombinant CD73 was able to induce T cell suppression of CD73⁻ CD4con T cells at a concentration as low as 0.15 ng/mL of the enzyme when the substrate AMP was provided (Fig. 2d). IFNγ production followed the same pattern as CD25 expression and proliferation in the

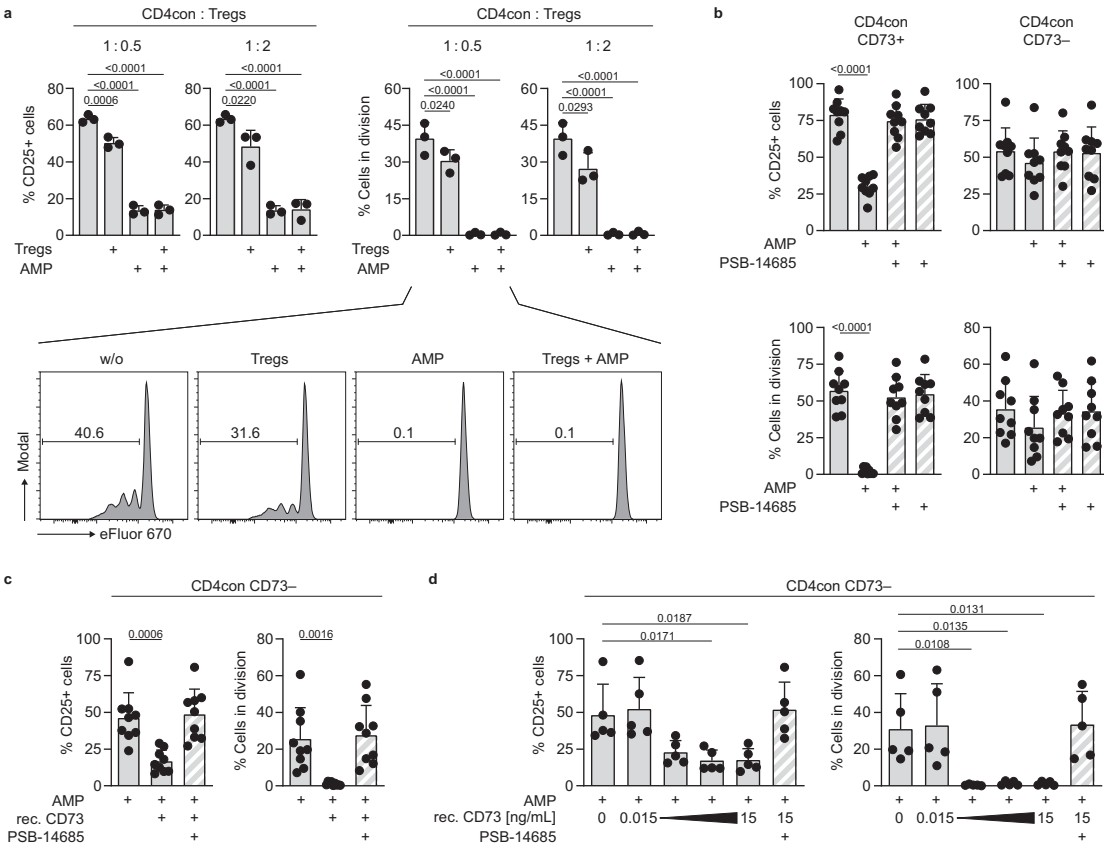

**Fig. 2 Treg-derived CD73 is not essential for adenosine-mediated suppression of conventional CD4 T cells. a–d** CD4 responder T cells were stimulated with αCD3/αCD28 in the presence of the ADA inhibitor EHNA (10 μM). CD25 expression and proliferation were measured after 4 days by flow cytometry. **a** CD4con T cells were stimulated and incubated with AMP (50 μM) and Tregs in the indicated ratio. **b** CD4con T cells were sorted into CD73⁻ and CD73⁺, the cells were incubated with AMP (50 μM) and the specific CD73 inhibitor PSB-14685 (10 μM). **c** CD4con CD73⁻ T cells were incubated with AMP (50 μM), the specific CD73 inhibitor PSB-14685 (10 μM), and recombinant CD73 (15 ng/mL). **d** CD4con CD73⁻ T cells were incubated with AMP (50 μM) and different concentrations of soluble recombinant CD73 (three tenfold serial dilutions starting with 15 ng/mL). PSB-14685 was used to block the highest concentration of recombinant CD73. Data were shown for **a** three, **b**, **c** nine and **d** five donors (mean ± SD). Ordinary one-way ANOVA with Dunnett's multiple comparisons test was used to compare all conditions to cells treated with EHNA or EHNA and AMP (first bar).

experiments shown in Fig. 2a–d (Supplementary Fig. 3a–d). We conclude that CD73-mediated production of adenosine constitutes an intrinsic mechanism of conventional T cells to control ongoing activation.

**Enzymatically active CD73 is released from the T cell membrane upon activation.** In human peripheral blood (PB) T cells, CD73 is predominantly expressed on CD8 T cells (Fig. 1f), and we have previously shown that activated T cells lose the membrane expression of CD73[31]. We hypothesized that the CD73 released from activated cells plays a role in adenosine generation. To address this point, we first investigated the time point of release in peripheral blood mononuclear cells (PBMCs) during activation. We observed a peak of CD73 expression on CD8 T cells 1 day after stimulation, followed by a marked decrease 2 or 3 days after stimulation (Fig. 3a). The loss of CD73 from the cell membrane was also detected in CD4 T cells. CD39 was upregulated in both T cell populations. Even though the loss of CD73 from the cell membrane was a general phenomenon, we observed interindividual variability both in the timing and extend of CD73 loss, and also in the level of CD39 upregulation.

Next, we investigated the fate of the released form of CD73. CD73 is a GPI-anchored protein that can be found as an enzymatically active soluble form, and CD73-specific AMPase activity is present in human plasma[32,33]. We implemented a sensitive, HPLC-based assay

with improved signal-to-noise sensitivity using fluorescent 1,$N^6$-etheno-AMP (eAMP) as a substrate to measure AMPase activity in cells and cell culture supernatants (Fig. 3b). As a proof of principle, we forced shedding of cell surface CD73 on CD8 T cells using phosphatidylinositol-specific phospholipase C (PI-PLC) (Fig. 3c, left) and observed lower AMPase activity of the PI-PLC-treated cells compared to untreated cells (Fig. 3c, middle). In contrast, the supernatants from PI-PLC treated cells converted more eAMP to 1,$N^6$-etheno-adenosine (eADO), in line with a higher amount of CD73 present in the supernatant (Fig. 3c, right). The CD73-specific inhibitor PSB-14685 completely blocked eAMP degradation, verifying the specificity of the AMPase activity. Activated T cells that had lost CD73 expression (Fig. 3d, left), showed lower AMPase activity than unstimulated cells (Fig. 3d, middle). This decrease in activity in the cell-bound compartment was paralleled by increased generation of eADO in the cell culture supernatant of activated cells and accompanied by an increase in the concentration of soluble CD73 detected by ELISA (Fig. 3d, right). These data demonstrate that CD73 is released in the cell culture supernatant upon T cell activation and retains its enzymatic activity.

**EVs are the major source of AMPase activity in CD8 T cell culture supernatants and mediate suppression of responder T cells.** We next sought to decipher the mechanism of CD73 release from the membrane. We first considered enzyme-

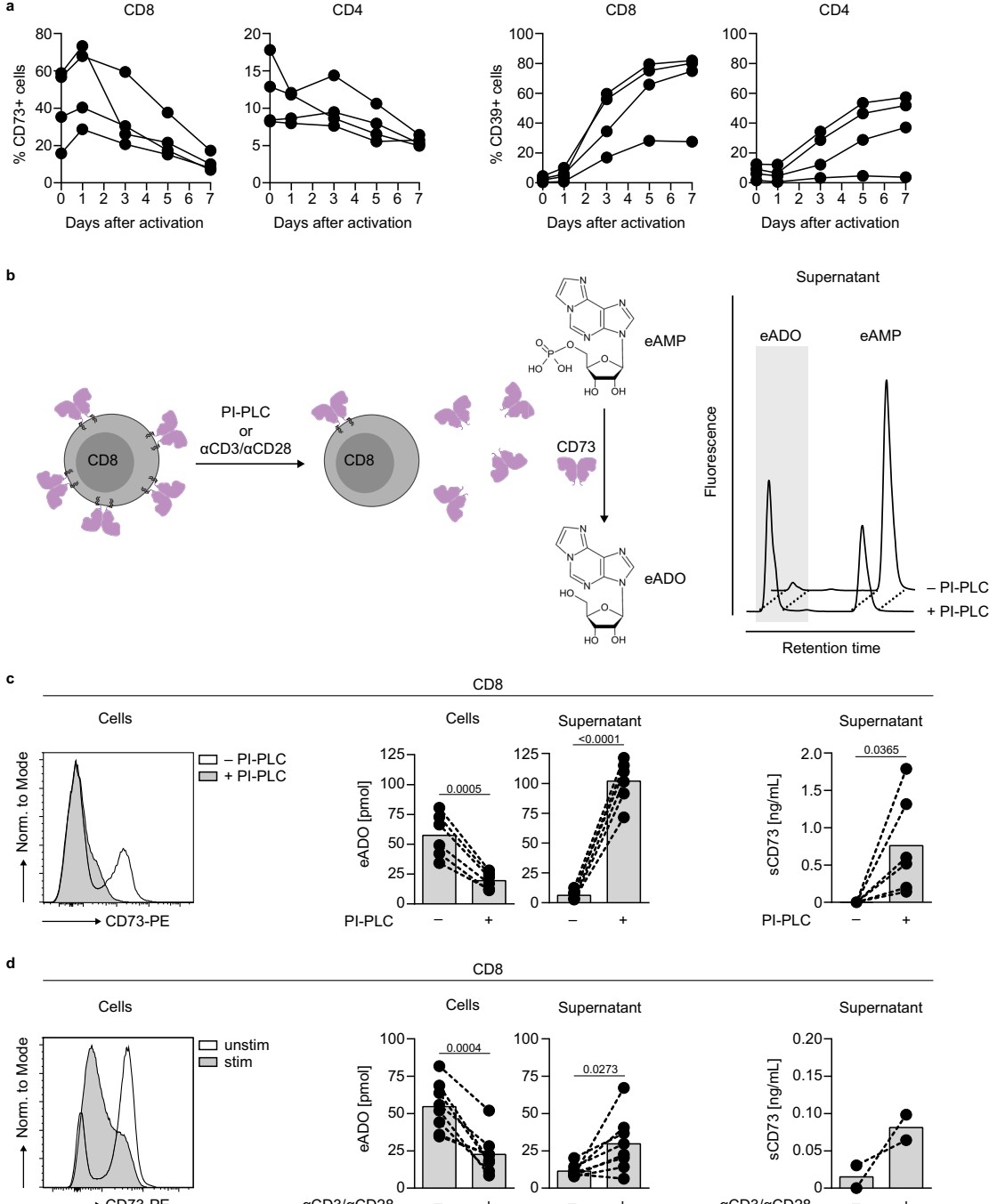

**Fig. 3 CD8 T cells release enzymatically active CD73 upon activation. a** Flow cytometric analysis of CD73 and CD39 expression on CD8 and CD4 T cells after activation of PBMCs (measurement on five time points from day 0 to day 7). Data were shown for four donors. **b** Schematic representation of CD73 released from CD8 T cells after PI-PLC treatment or activation. AMPase activity is measured by the conversion of $1,N^6$-etheno-AMP (eAMP) to $1,N^6$-etheno-ADO (eADO). **c, d** Determination of the AMPase activity of CD8 T cells and supernatants. $0.2 \times 10^6$ CD8 T cells were treated with **c** PI-PLC (0.5 U/mL) or **d** stimulated with αCD3/αCD28 for 4 days. Histograms show CD73 expression of the cells at the time of eAMP incubation for one selected donor. The cells and supernatants were incubated with eAMP and degradation of eAMP to eADO was determined by HPLC (middle panel, mean of **c** six donors and **d** eight donors). The amount of soluble CD73 (sCD73) was measured by ELISA (right panel, mean of **c** six donors and **d** two donors). A two-tailed paired *t*-test was used to compare untreated and treated samples in **c, d**.

mediated shedding; however, neither specific inhibition of phospholipase D nor inhibition of metalloproteinases prevented loss of CD73 from the cell surface (Supplementary Fig. 4). CD73 is enriched in lipid rafts of T cells after activation, suggesting that the release might occur in form of vesicles, as it has been described in tumor cells[34]. Using differential centrifugation of cell culture supernatants, we observed a stepwise reduction in the

AMPase activity in the supernatants of stimulated CD8 T cells after $10,000 \times g$ (large vesicles and apoptotic bodies are pelleted) and $110,000 \times g$ centrifugation (small vesicles are pelleted), indicating that CD73 is, indeed, present on vesicles (Fig. 4a). To verify the specificity of the CD73-mediated enzymatic activity on these vesicles, supernatants from activated $CD73^-$ and $CD73^+$-sorted CD8 T cells were subjected to differential centrifugation and

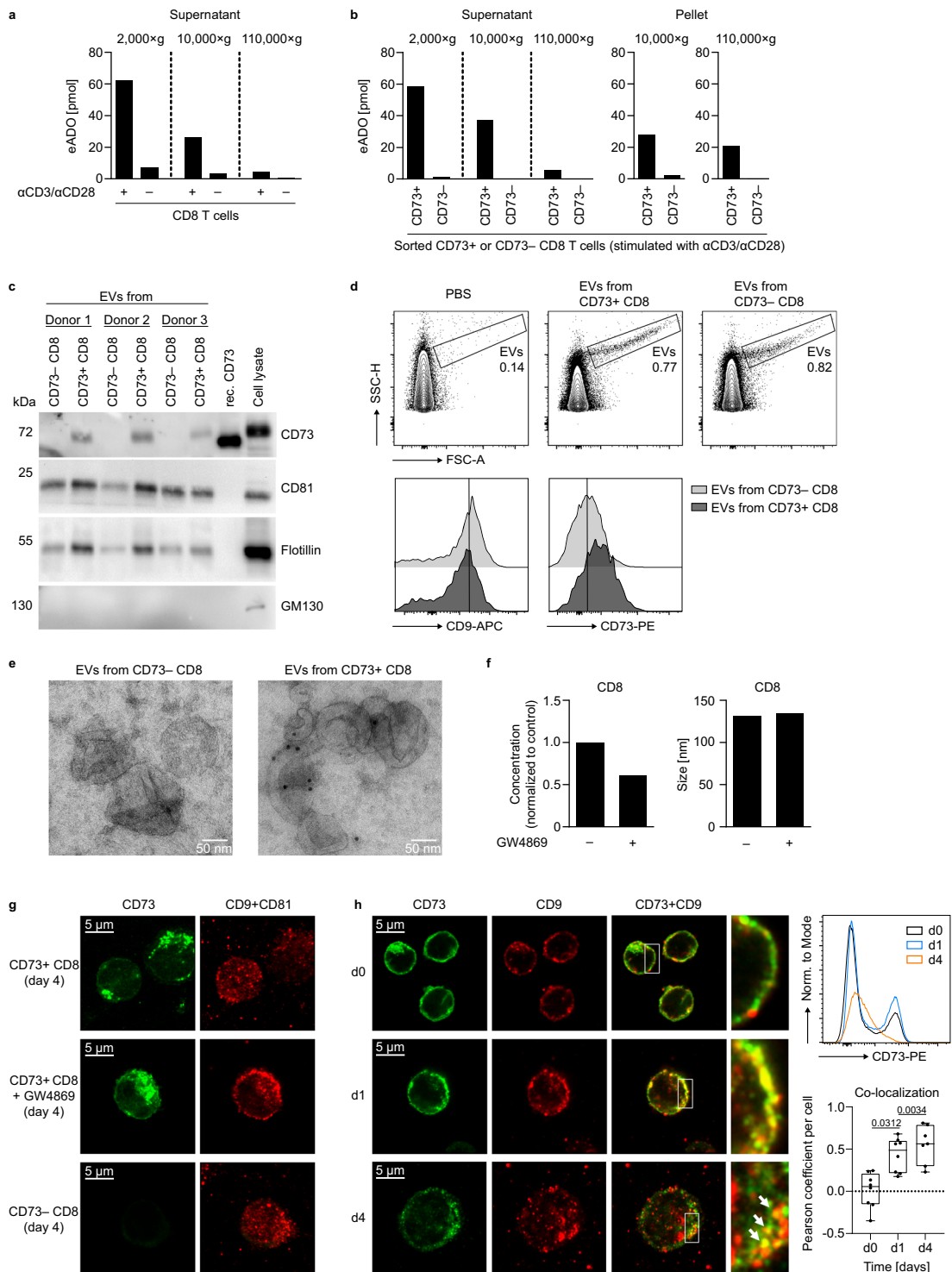

AMPase activity was measured in the obtained fractions. The cell culture supernatants of CD73+ CD8 T cells displayed AMPase activity after $2000 \times g$ centrifugation (removal of cell debris), but ultracentrifugation heavily reduced eADO generation. Remarkably, the EV-enriched pellets derived from $110,000 \times g$ centrifugation (hereinafter named EVs) showed substantial AMPase activity (Fig. 4b). Supernatants from CD73− CD8 T cells did not have AMPase activity, demonstrating the specificity of CD73 for the enzymatic degradation of AMP in these samples.

Nanoparticle tracking analysis (NTA) of CD8 T cell EVs revealed a size of around 100 nm for the majority of the isolated CD8 T cell vesicles (Supplementary Fig. 5). The presence of EV markers CD81 and flotillin, as well as the absence of the Golgi protein GM130 confirmed the EV nature and purity of our samples. In addition, CD73 protein was detected in EVs derived from CD73+ CD8 T cells, but not in those from CD73− CD8 T cells (Fig. 4c). Even though the small size of the EVs limits precise analysis by conventional flow cytometry, particles gated by forward vs. side scatter showed increased staining intensity of CD73 in vesicles derived from CD73+ CD8 T cells compared to those derived from CD73− CD8 T cells, while expression of tetraspanin CD9 was comparable (Fig. 4d). Electron microscopy

**Fig. 4 Extracellular vesicles are the major source of AMPase activity in human CD8 T cell culture supernatants. a**, **b** CD8 T cells were stimulated with αCD3/αCD28 and kept in culture for 4 days. Analysis of AMPase activity in **a** cell culture supernatants of stimulated or unstimulated total CD8 T cells and **b** supernatants and EVs of sorted CD8 CD73⁺ or CD8 CD73⁻ T cells after differential centrifugation. The conversion of eAMP to eADO was measured by HPLC. **c**–**e** Analysis of CD73 and EV markers on EVs derived from stimulated cell culture supernatants of CD73⁺ or CD73⁻ CD8 T cells by **c** western blot, **d** flow cytometry, and **e** electron microscopy with immunogold labeling. Recombinant CD73 or cell lysate from stimulated CD8 T cells served as positive controls. **f** EVs were isolated from CD8 T cells stimulated with αCD3/αCD28 for 3 days in the presence or absence of GW4869 (10 μM), and particle concentration and size were measured by NTA. **g** Microscopy analysis of CD73, CD9, and CD81 expression in sorted CD73⁺ or CD73⁻ CD8 T cells after 4 days of stimulation with αCD3/αCD28. GW4869 (10 μM) was added at day 0. **h** Microscopy analysis of CD73 and CD9 expression in CD8 T cells before and after activation. Pearson coefficient was determined to quantify the co-localization of CD73 and CD9. Data were analyzed from seven (d4) or eight (d0, d1) high power fields (center line: median, box limits: 25th to 75th percentiles, whiskers: min to max). Donors analyzed: **a**, **b** three donors in three independent experiments; **c** six donors in three independent experiments; **d**–**f** three donors; **g**, **h** samples from three donors processed independently. Each figure panel shows a representative donor/experiment. Kruskal–Wallis test with Dunn's multiple comparisons test was used to compare Pearson coefficients of co-localization in **h**.

of the purified vesicles revealed the characteristic cup-shaped morphology with sizes between 50 to 200 nm. Immunogold labeling further confirmed the presence of CD73 in EVs derived from CD73⁺ CD8 T cells (Fig. 4e). To investigate the vesicular release of CD73, we treated T cells with GW4869, a pharmacological compound that blocks the generation of a specific type of EVs called exosomes[35]. The addition of GW4869 to the cell culture reduced the amount of generated EVs whereas the size was not affected (Fig. 4f). It also prevented the loss of tetraspanins CD9 and CD81 from the cell (Fig. 4g). High-resolution fluorescence microscopy of CD8 T cells showed the reduction of CD73 surface expression after 4 days of stimulation (Fig. 4h) consistent with the results obtained by flow cytometry (Fig. 4h upper right panel and Fig. 3a). Overlay of CD73 with tetraspanins CD9 and CD81 revealed significant co-localization of CD73 with vesicle markers at day 1 and day 4 after activation (Fig. 4h lower right panel and Supplementary Fig. 6). Altogether, these data show that CD73 is released with EVs upon T cell activation and that these EVs comprise the majority of AMPase activity in T cell culture supernatants.

We wondered if CD73-containing EVs isolated from T cell culture supernatants functionally compensate the lack of membrane-bound CD73 in the control of T cell activation and proliferation as we have shown for recombinant CD73 (Fig. 2c). To test if EVs, a natural source of non-cell-bound CD73, can suppress T cell function, we activated CD73⁻ CD4con T cells (to prevent any effect of residual membrane-bound CD73 on the responder T cells) in the presence of AMP. Addition of recombinant CD73 or EVs isolated from the supernatant of CD73⁺ CD8 T cells induced strong suppression, which was dose-dependent and could be blocked by the CD73 inhibitor PSB-14685 (Fig. 5a, b and Supplementary Fig. 7). Importantly, EVs isolated from the supernatant of CD73⁻ CD8 T cells did not suppress activation and proliferation of the responder T cells, and neither did the pelleted material after ultracentrifugation of cell culture medium (Fig. 5 and Supplementary Fig. 8), indicating that the observed suppressive effect is CD73-specific and not due to contaminants in the EV preparation. In summary, we show that enzymatically active CD73 is released from activated CD8 T cells with EVs, and that EVs derived from activated CD73⁺ T cells are highly suppressive.

**T cell-derived EVs released during immune cell activation cooperate with regulatory T cells for the efficient suppression of effector T cells.** In our previous experiments, we exogenously provided AMP, the substrate for CD73. AMP is generated from ATP by the enzymatic activity of CD39. In contrast to conventional T cells, a substantial proportion of Tregs constitutively express CD39 in humans (Fig. 1f). Therefore, Tregs are likely to be superior at degrading ATP than conventional T cells. We used

the sensitive HPLC-based assay to compare the ATPase activity of purified Tregs, CD4con, and CD8 T cells after activation. Using 1,$N^6$-etheno-ATP (eATP) as substrate, we found that Tregs had the highest ATPase activity of all tested T cell subsets (Fig. 6a), and this was more pronounced in donors with substantial basal CD39 expression on Tregs (Fig. 6a, left panel) than in donors with low frequency of CD39-expressing Tregs (Fig. 6a, right panel). Notably, CD8 T cells were the second most efficient cell type in degrading ATP, while CD4con T cells showed the least ATPase activity, in agreement with the percentage of cells upregulating CD39 after activation (Fig. 3a).

We suspected that the high ATPase activity of Tregs combines with CD73-containing EVs to mediate maximum immune suppression. We performed a suppression assay, using CD73⁻ CD4con T cells as responder T cells, and added different ratios of Tregs. With a ratio of CD4con T cells to Tregs close to the physiological situation (1:0.125), Tregs could only induce partial suppression. The addition of EVs from CD73⁺ CD8 T cells, a natural source of non-cell-bound CD73, reduced CD25 expression, and completely suppressed proliferation and IFNγ production of effector T cells (Fig. 6b). Importantly, EVs derived from CD73⁻ CD8 T cells had no additional effect on the suppression of responder T cells, revealing that they do not cooperate with Tregs to generate immunosuppressive adenosine. We conclude that Tregs are the main providers of ATPase activity in our system, and the concerted action between Treg-derived ATPase activity and effector cell-derived AMPase activity in form of EVs ensures the best conditions for immune suppression.

**EVs isolated from the synovial fluid of patients with juvenile idiopathic arthritis are immunosuppressive.** An interesting question to address is the relevance of non-cell-bound CD73 in the context of inflammation. The SF of patients with autoimmune arthritis offers an ideal chance to analyze an inflammatory compartment in the human system because we can access disease-relevant infiltrating immune cells. Flow cytometric analyses of the cellular composition of the SF from eight children with JIA (Fig. 7a) revealed that T cells constitute the main cell population, and that CD8 T cells are enriched in the SF compared to PB in most cases. B cells, the only other prominent CD73-expressing cell type in the human immune compartment[2,28], are clearly underrepresented in the SF compared to PB in all patients. We next compared the expression of CD73 and CD39 on the T cells from PB and SF (Fig. 7b). In agreement with published data[36,37], we found increased expression of CD39 on CD4 T cells and Tregs and decreased CD73 expression on CD8 T cells in the synovial T cell subsets compared to their peripheral counterparts. In analogy to our in vitro findings after activation, we suspected that CD73 has been lost from the cell membrane. Indeed, we detected moderate to high concentrations of soluble CD73 in the

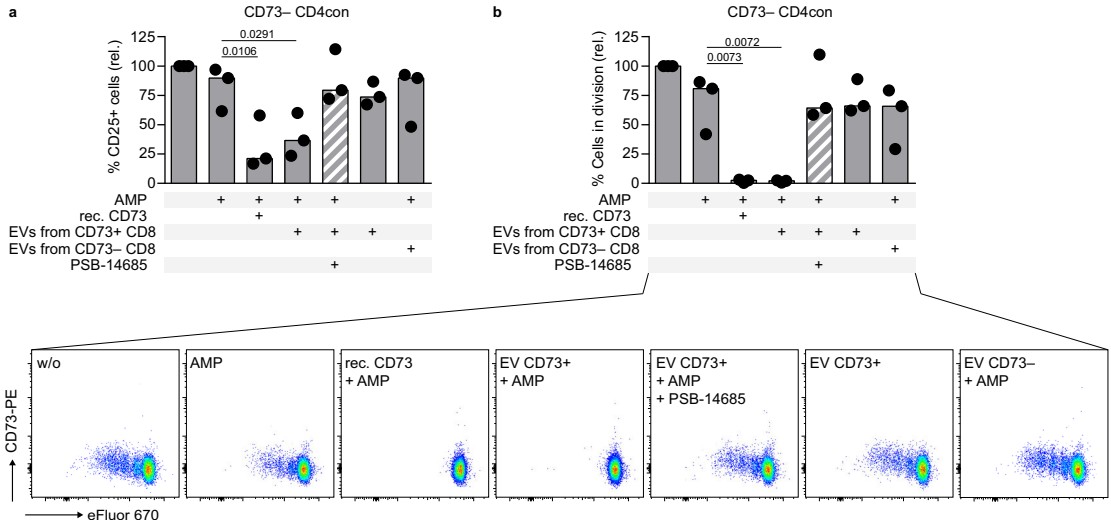

**Fig. 5 Extracellular vesicles isolated from the cell culture supernatant of activated CD73+ CD8 T cells are immunosuppressive.** CD73− CD4con T cells were stimulated with αCD3/αCD28 in the presence of the ADA inhibitor EHNA (10 μM) and incubated with AMP (50 μM), recombinant CD73 (15 ng/mL), EVs and PSB-14685 (10 μM) as indicated. **a** CD25 expression and **b** proliferation were measured after 4 days by flow cytometry. Data were shown as a median of three donors from independent experiments. For each donor, CD25 expression and proliferation were set in relation to cells only treated with EHNA. Repeated measures one-way ANOVA with Dunnett's multiple comparisons test was used to compare all conditions to cells treated with EHNA and AMP (second bar).

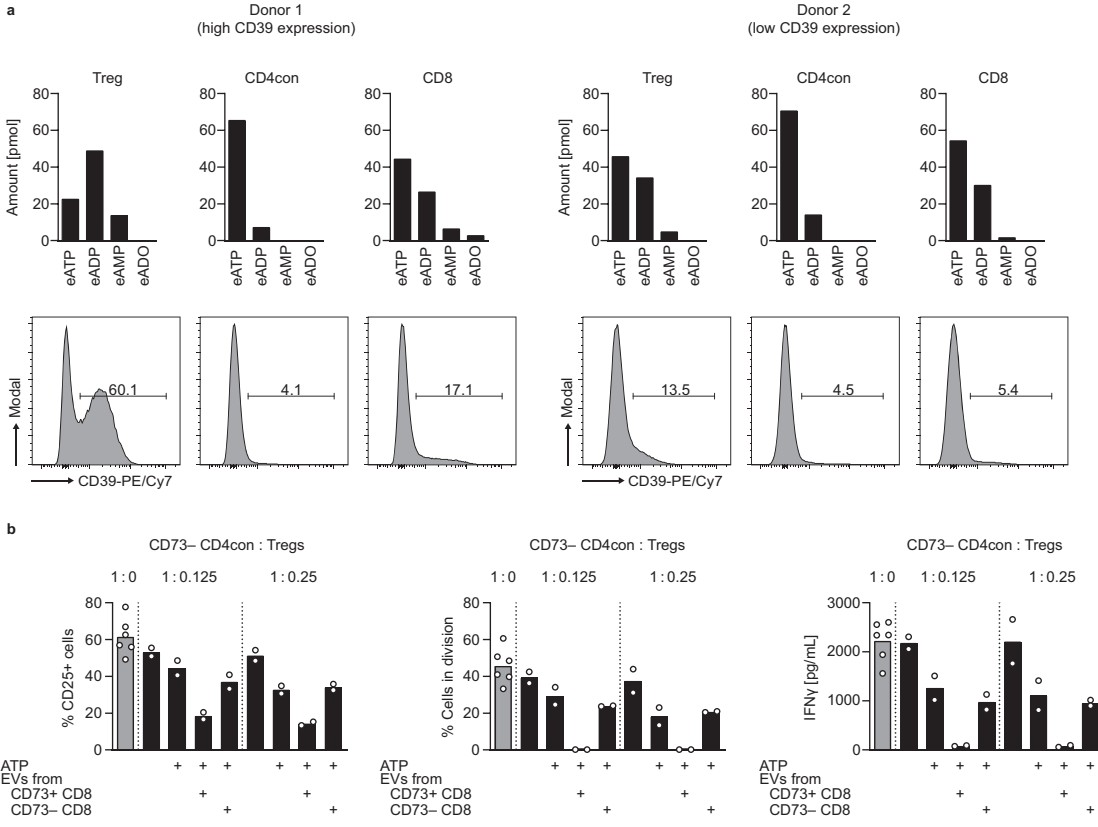

**Fig. 6 The combined activities of Treg-derived CD39 and vesicular CD73 result in optimal suppression of T cell function. a** Analysis of ATPase activity in sorted T cell subsets of two donors by HPLC. Activated cells (αCD3/αCD28, 3 days) were incubated with 1,$N^6$-etheno-ATP (eATP) and degradation of etheno-nucleotides was determined by HPLC. Histograms show CD39 expression on the cells at the time of eATP incubation. **b** CD73− CD4con T cells were stimulated with αCD3/αCD28 in the presence of the ADA inhibitor EHNA (10 μM) and incubated with Tregs, ATP (50 μM), and EVs from cell culture supernatants of sorted and activated CD8 T cells as indicated. CD25 expression and proliferation were measured after 4 days by flow cytometry. IFNγ production was determined by ELISA in the cell culture supernatant harvested on day 4. Data were shown for one representative donor out of four analyzed (mean of technical replicates).

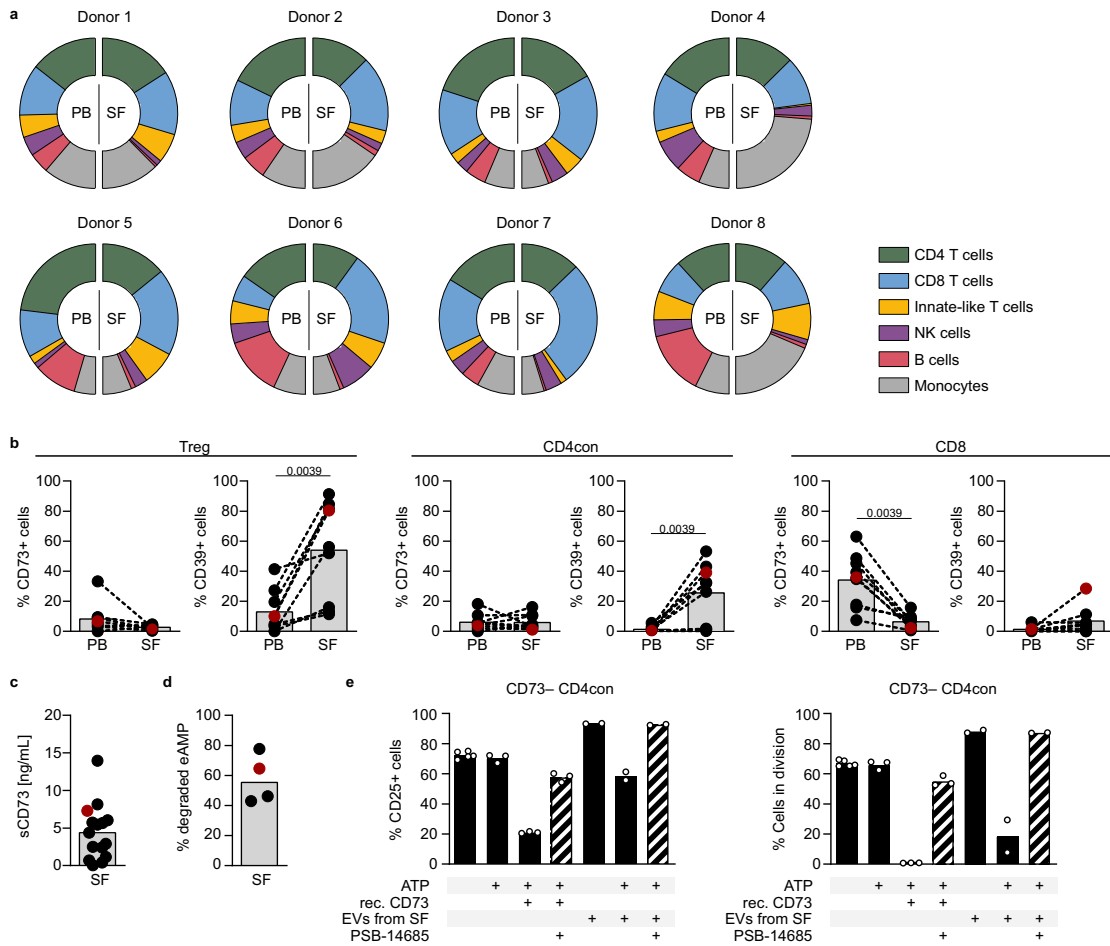

**Fig. 7 Extracellular vesicles isolated from synovial fluid of patients with juvenile idiopathic arthritis are immunosuppressive. a, b** Flow cytometric analysis of **a** immune cell subsets and **b** CD73 and CD39 expression in SF and PB of JIA patients. **c** CD73 in the cell-free moiety of SF was measured by ELISA. **d** SF was incubated with eAMP and degradation of eAMP was determined by HPLC. **e** CD73⁻ CD4con T cells were stimulated with αCD3/αCD28 in the presence of the ADA inhibitor EHNA (10 µM) and incubated with ATP (50 µM), recombinant CD73 (15 ng/mL), EVs (3.1 × 10⁸ particles), and PSB-14685 (10 µM) as indicated. CD25 expression and proliferation were measured after 4 days by flow cytometry. Data were shown for **a** eight individual donors, as mean of **b** nine, **c** fifteen, and **d** four donors indicated by dots, or for **e** a representative donor out of three analyzed (mean of technical replicates), marked as a red dot in **b–d**. A two-tailed Wilcoxon test was used to compare ectonucleotidase expression in PB and SF in **b**.

SF of nearly all patients tested (Fig. 7c) and confirmed AMPase activity in the samples measured (Fig. 7d). We next isolated EVs from the SF and verified their EV nature by electron microscopy and western blot. The EV markers flotillin and CD81 could be detected in all EV samples along with CD73 (Supplementary Fig. 9a, b). We also assessed for the presence of protein contamination and found rests of albumin, as previously reported when using differential ultracentrifugation for EV isolation[38], but not of apolipoproteins (Supplementary Fig. 8b).

Among other cell types, T cells clearly contributed to the cellular sources of EVs in the SF (Supplementary Fig. 9c), and we could show co-expression of CD8 and CD73 on these EVs by conventional flow cytometry (Supplementary Fig. 9d). We assessed if SF-derived EVs suppress CD73⁻ CD4con T cells. As observed before, the addition of recombinant CD73 and ATP to the responder T cells resulted in reduced T cell activation and completely abolished proliferation. When SF-derived EVs were added to the culture in the absence of ATP, we observed an increase in activation and cell proliferation. Importantly, in the presence of ATP, this increase was abrogated, and a clear reduction in cell proliferation was induced, which was reversed by the addition of the CD73-specific inhibitor PSB-14685 (Fig. 7e). Of note, SF-derived EVs decreased the activation and

proliferation of responder T cells in a dose-dependent manner (Supplementary Fig. 9e). In summary, we show here that EVs from the SF of JIA patients degrade AMP, generating adenosine and suppressing T cell proliferation and function.

## Discussion

Adenosine is a potent regulator of inflammation generated in the extracellular space by the sequential hydrolysis of ATP by ecto-nucleotidases CD39 and CD73. We show here that activated human CD8 T cells release CD73-containing EVs that generate adenosine, contributing substantially to immune suppression.

The degradation of ATP and generation of adenosine is a well-described mechanism of suppression in murine Foxp3⁺ Tregs, which express both CD39 and CD73 on their cell surface[14]. However, very few human FOXP3⁺ Tregs express CD73, and they are not particularly adept at adenosine generation. We describe here how CD73-mediated generation of adenosine is mostly independent from Tregs in the human system, and propose a coordinated effort involving CD39 on Tregs for the degradation of ATP to AMP, and CD73 on T cell-derived EVs that provide the necessary AMPase activity to generate adenosine. We have shown the release of CD73-containing EVs from activated CD8 T cells. However, CD4 T cells also lose CD73 from the

cell membrane upon activation, suggesting a parallel mechanism for CD4 T cells.

Intracellular ATP functions as an essential energy source. Once released into the extracellular space, ATP becomes a biologically active signaling molecule[39]. Conditions such as cellular activation and stress, inflammation, ischemia, or hypoxia promote ATP release and raise the pericellular concentration of ATP sufficiently to activate P2 receptors and support inflammation[25,40,41]. Extracellular ATP is rapidly hydrolyzed by ectonucleotidases in a stepwise manner to yield adenosine, a potent immune suppressor upon binding to P1 receptors $A_{2A}$ and $A_{2B}$ on immune cells. The degradation of adenosine by ADA further regulates the availability of adenosine for signaling. Thus, a complex network of purinergic enzymes and receptors controls the duration and magnitude of purinergic signaling and regulates the immune response[42]. Remarkably, the affinity of purinergic molecules for its receptors and the outcome after receptor activation are cell- and species-specific[43]. Adenosine signaling supports the expansion and suppressive function of murine Tregs[15] while inhibiting effector T cells. In humans, adenosine inhibits both Tregs and effector T cells[44]. Not only the response to extracellular adenine nucleotides is different between mice and humans, but also the expression and regulation of ectonucleotidases CD39 and CD73. The co-expression of both enzymes on murine Tregs secures the production of adenosine, and this metabolic path is important for their suppressive function[14,45]. In contrast, very few circulating human Tregs express CD73 and they are still suppressive. This poses the question of whether human Tregs rely on adenosine generation or rather favor other mechanisms of immune suppression. The absence of CD73 on the membrane of human Tregs could be an evolutionary advantage to avoid the production of pericellular adenosine that inhibits their function[44]. Further, human Tregs do not express CD26, a docking site for ADA at the cell membrane[46], lacking a mechanism for efficient degradation of pericellular adenosine.

CD73, as a GPI-anchored protein, clusters at the plasma membrane in lipid rafts that are poised for EV formation[47,48]. We found that CD73 diminished drastically from the surface of T cells 3 to 4 days after activation, concomitant to the loss of AMPase activity in the cellular compartment. In parallel, enzymatic activity increased in the cell culture supernatants. Importantly, the capacity to produce adenosine found in the cell culture supernatants was lost after ultracentrifugation, indicating that it was contained in EVs and not present as a soluble protein. Part of the AMPase activity was already lost after $10,000 \times g$ centrifugation, which we attribute to CD73 on apoptotic bodies or larger EVs. Even though CD73 has higher enzymatic activity in its soluble form compared to the membrane-bound variant[49], we speculate that the vesicular form has advantages, such as an extended half-life, and better distribution through body fluids. It is not yet known how far EVs can travel in human body fluids, but murine EVs can rapidly traffic to the spleen and liver before elimination 6 h after intravenous injection[50]. The generation of EVs poses the question of how the cells recover the membrane loss. A possibility would be the fusion with foreign EVs, which would permit a very dynamic exchange of membrane components between cells and the acquisition of new functions[51,52].

The exact contribution of EV- and T cell-derived adenosine to immune suppression in vivo is difficult to predict due to the multiple levels of regulation that influence the availability of adenine nucleotides during an immune response. Contributing factors include the rapid generation and degradation of signaling-relevant molecules, local differences in nucleotide concentrations, potential feed-forward inhibition mechanisms, and alternative degrading enzymes. In our in vitro experiments with human T cells, we used different substrates and specific inhibitors to overcome these difficulties. By blocking the degradation of adenosine with an ADA inhibitor, we emphasized the role of the purinergic cascade in immune suppression. The addition of exogenous ATP at a moderate concentration (50 μM) served to mimic local increases of ATP as it has been reported in different settings of inflammation and ischemia[53]. In the absence of Tregs, we added AMP to ensure the availability of enough substrate for CD73, because conventional T cells are much less efficient than Tregs in producing AMP. Furthermore, adding AMP as substrate bypasses a potential feed-forward inhibition of CD73 by ATP or ADP[54]. Although CD73 is the main AMP-degrading enzyme, adenosine is still generated by tissue-nonspecific alkaline phosphatase (TNAP) when CD73 is absent, as shown in CD73 knockout mice[55,56] and in cells from patients with CD73 deficiency[57]. In our experiments, we could completely abolish AMPase activity with a specific CD73 inhibitor[58,59], indicating that TNAP activity is negligible in our system. The observed ATPase activity is likely CD39-mediated, but we cannot exclude a contribution of other enzymes, such as ENPP1 (ectonucleotide pyrophosphatase/phosphodiesterase-1) or other members of this family of pyrophosphatases[60]. Of note, the expression, regulation, and activity of ENPP1 on T cells has not been explored. The detection of ADP as an intermediate product in our assays indicates a stepwise degradation of ATP, favoring the role of ectonucleoside triphosphate diphosphohydrolases (ENTPDases) like CD39, but not pyrophosphatases.

Both CD39 and CD73 are expressed in distinct immune cell types. In humans, apart from T cells, B cells express both ectonucleotidases, while monocytes and dendritic cells express mostly CD39. Also, endothelial cells and mesenchymal cells express CD73, and all these cell types contribute to ATP metabolism in vivo. EVs containing CD39 or CD73 have been isolated from mesenchymal stem cells, B cells, and Tregs[24,61,62]. These EVs have immunoregulatory properties[63] and their modulatory role has been reported in cancer[34,62]. We have not addressed here the contribution of other ectonucleotidases present in T cell-derived EVs to ATP metabolism, but it has been shown that Treg-derived EVs contain CD39 and inhibit T cell proliferation[24]. Further work will be required to dissect the role of cellular and EV-associated ectonucleotidases in immune suppression.

We had the unique chance to isolate EVs from a human site of inflammation, the SF of JIA patients. The SF contains predominantly monocytes and memory T cells that produce inflammatory cytokines, sustaining inflammation in the affected joints[64,65]. In agreement with previous reports, we found increased CD39 and negligible expression of CD73 in the T cells of the SF. While SF-infiltrating cells show ATPase activity, AMPase activity is decreased in the cellular fraction[36,37]. Our analysis revealed that CD73 is present in the cell-free fraction of the SF, including EVs. The cellular source for these EVs are T cells, monocytes/macrophages, and endothelial and mesenchymal stem cells. Because human monocytes do not express CD73[26,27], we concluded that activated T cells are a relevant source of vesicular CD73. This was confirmed by the co-expression of CD73 and CD8 on SF-derived EVs. The endothelial cells in the synovial lining and mesenchymal stem cells, both expressing CD73, are also plausible sources for CD73+ SF EVs. This finding deserves further investigation. Our in vitro experiments showed that SF-derived EVs promote T cell activation and proliferation, probably due to cytokines and growth factors contained in these EVs[66]. However, by strengthening the purinergic signaling cascade with the addition of ADA inhibitor EHNA and ATP, we turned the outcome around, and EVs became immunosuppressive, as it has been shown for CD73-containing EVs produced by tumor cells[34]. Therefore, in an ATP-rich environment, as it is at sites of inflammation and ischemia or

in the intestine, the purinergic signaling cascade becomes highly relevant for immune regulation[67].

We conclude that abundant CD39 expression on Tregs secures the hydrolysis of ATP to ADP and AMP, but that the AMPase activity of CD73 is provided by CD8 T cells, and mostly contained in EVs. We propose that adenosine generation by conventional (nonregulatory) T cells is a built-in mechanism of immune suppression necessary to restrain ongoing inflammation, contesting the common paradigm that Tregs must provide the whole machinery for adenosine production. Finally, our results highlight the role of EVs in the control of immune responses and support the prospect of modulating the purinergic axis for the treatment of local inflammation.

## Methods

**Origin of samples and isolation of human mononuclear cells**. Buffy coats were obtained from the blood bank of the University Medical Center Hamburg-Eppendorf (UKE). PB was drawn from healthy volunteers visiting the UKE. Blood and SF of patients with JIA, specifically oligo- and polyarthritis, were obtained from children visiting the UKE, the Altona Children's Hospital, the University Medical Center Schleswig-Holstein (campus Lübeck), or the Medical Center Bad Bramstedt. SF was obtained from joint puncture for diagnostic or therapeutic reasons. All samples were handled according to corresponding ethics protocols (Ethics Committee of the Hamburg Chamber of Physicians, protocols PV5139 for samples from healthy donors, and PV3746 for samples from JIA patients), and informed consent was obtained from all donors. Mononuclear cells (MC) were isolated from blood and SF by Biocoll density gradient centrifugation (Merck). Isolated immune cells were used for flow cytometric analyses or further isolation of T cell subsets.

**Isolation of murine splenocytes**. All mouse experiments were performed in accordance with national and institutional guidelines on animal care (Hamburg Authority for Health and Consumer Protection, Veterinary Affairs/Food Safety, protocol ORG983). Lymphocytes were isolated from the spleen of C57BL/6 mice housed in the animal facility at the UKE. Single-cell suspensions were prepared by processing the spleen through a 70 μm strainer. After erythrocyte lysis, splenocytes were used for the analysis of surface markers by flow cytometry.

**Flow cytometry and fluorescence-activated cell sorting (FACS)**. Human and murine immune cells were preincubated with immunoglobulins to block unspecific binding and stained with fluorescence-labeled antibodies for 30 min at 4 °C. All antibodies were titrated prior to use. The following fluorochrome-conjugated anti-human antibodies were used: anti-CD3 (clone UCHT1 and clone OKT3), anti-CD4 (clone RPA-T4, clone SK3, and clone OKT4), anti-CD8α (clone RPA-T8 and clone HIT8a), anti-CD9 (clone HI9a), anti-CD16 (clone 3G8), anti-CD19 (clone HIB19), anti-CD25 (clone BC96), anti-CD39 (clone A1), anti-CD73 (clone AD2), anti-CD127 (clone HCD127 and clone A019D5) (all BioLegend), anti-CD14 (clone M5E2), anti-CD25 (clone 2A3), anti-CD45 (clone HI30), anti-TCRγδ (clone 11F2) (all BD Biosciences), anti-CD56 (clone N901) (Beckman Coulter). The following fluorochrome-conjugated anti-mouse antibodies were used: anti-CD3 (clone 17A2), anti-CD4 (clone RM4-5), anti-CD25 (clone PC61), anti-CD73 (clone TY/11.8) (all BioLegend), anti-CD8 (clone 53-6.7), anti-CD19 (clone 1D3), and anti-CD39 (clone 24DMS1) (all eBioscience). For dead cell exclusion, a live/dead dye (Thermo Fisher Scientific) was included. The staining cocktails were designed to minimize the effects of spectral overlap. Prior to analysis, a compensation matrix was calculated after single color staining of human PBMCs, as described[68]. Samples were measured at FACSCanto II, FACSCelesta, or LSR-Fortessa (BD Biosciences) using FACSDiva software for data acquisition (BD Biosciences) and analyzed using FlowJo software (BD). For isolation of specific T cell subpopulations by FACS, cells were stained as described above and sorted at FACSAria IIIU (BD Biosciences).

**Preparation and stimulation of human T cells**. CD8 or CD4 T cells were isolated from PBMCs by negative selection using the EasySep Human T Cell Enrichment Kit (Stemcell Technologies). The purity of the isolated cells was assessed by flow cytometry. For indicated assays, Tregs (defined as CD4+ CD25high CD127low) and conventional CD4 T cells (CD4con, defined as non-Treg CD4 T cells) were obtained by FACS. CD4con T cells were additionally sorted with regard to their CD73 expression if indicated. To assess proliferation in T cell assays, responder cells were labeled with 2 μM eFluor 670 (Thermo Fisher Scientific).

Purified T cells were stimulated with 1 μg/mL coated αCD3 (OKT3) and 5 μg/mL soluble αCD28 (CD28.2) (both BioLegend) and cultured in serum-free X-VIVO 15 medium (Lonza). PBMCs were activated with 0.5 μg/mL soluble αCD3 in RPMI containing 1% penicillin-streptomycin, 1% L-glutamine (all Thermo Fisher Scientific), and 10% FBS (Biochrom). If not otherwise stated, cells were seeded at a density of $1 \times 10^6$ cells/mL.

**Isolation of EVs**. EVs were isolated from cell culture supernatants, X-VIVO 15 medium, and SF by differential centrifugation at 4 °C. Samples were centrifuged at $450 \times g$ for 5 min, followed by centrifugation at $2000 \times g$ for 10 min. After centrifugation at $10,000 \times g$ for 30 min, the supernatant was subjected to ultracentrifugation ($110,000 \times g$, 70 min, in an SW 60 Ti swinging-bucket rotor, Beckman Coulter). The EV-enriched pellet (hereinafter named EVs) was washed and resuspended in PBS. Particle concentration and size were determined using the NTA instrument NanoSight LM14 (Malvern Panalytical) equipped with a 638 nm laser and a Marlin F-033B IRF camera (Allied Vision Technologies), operated with NTA 3.0 software.

### Characterization of EVs

*Western blot*. The EVs were characterized according to the ISEV guidelines[69]. For western blot analysis, EVs were incubated with RIPA buffer (50 mM Tris-HCl pH 7.4, 150 mM NaCl, 1% NP40, 0.5% Na-deoxycholate, and 0.1% SDS) in the presence of protease and phosphatase inhibitors (Roche). The protein content of EVs was assessed with a Micro BCA Protein assay kit (Thermo Scientific). Samples (3.5 μg per EV sample, 15 ng recombinant CD73, 10 μg cell lysate, 2 μL pure SF, 15 μL X-VIVO 15 medium, or 15 μL of X-VIVO ultracentrifugation pellet corresponding to 2 mL X-VIVO 15 medium) were mixed with 4X loading buffer (250 mM Tris-HCl, 8% SDS, 40% glycerol, 20% β-mercaptoethanol, 0.008% Bromophenol Blue, pH 6.8), boiled for 5 min at 95 °C and subjected to electrophoresis on a 10% Bis-Tris gel (Invitrogen) under denaturing conditions. The gel was electroblotted onto a nitrocellulose membrane (LI-COR) and stained with the Revert Total Protein Stain (LI-COR). After blocking for 60 min with Roti-Block (Carl Roth), membranes were incubated overnight with the following primary antibodies diluted in Roti-Block: anti-CD73 (clone D7F9A, Cell Signaling, 1:1000), anti-CD81 (clone D3N2D, Cell Signaling, 1:1000), anti-Flotillin-1 (clone 18/Flotillin-1, BD Biosciences, 1:1000), anti-GM130 (clone 35/GM130, BD Biosciences, 1:500), anti-Albumin (clone F-10, Santa Cruz, 1:1000), anti-apoA-1 (clone E-20, Santa Cruz, 1:500), or anti-apoB (polyclonal, Acris, 1:1000). After washing with TBST, the membranes were incubated for 60 min with anti-rabbit-HRP-conjugated secondary antibody (Cell Signaling, 1:1000), or anti-mouse-HRP-conjugated secondary antibody (Cell Signaling, 1:1000), or anti-goat-HRP-conjugated secondary antibody (Jackson ImmunoResearch Laboratories, 1:5000). Membranes were developed with SuperSignal West Femto Maximum Sensitivity Substrate (Thermo Fisher Scientific), and images were taken on a ChemiDoc Imaging System (Bio-Rad) using the Quantity One software.

*Electron microscopy*. For electron microscopy with immunogold labeling, EVs were adsorbed to carbon-coated grids for 20 min, washed with PBS, quenched in glycine, and blocked with blocking solution for goat gold conjugates (Aurion)[70]. After 30 min incubation with the primary anti-CD73 antibody (clone AD2, BioLegend, 5 μg/mL), the grids were incubated for 20 min with the anti-mouse IgG 10 nm gold conjugate (Sigma, 1:20), fixed in 2.5% glutaraldehyde, washed with water, and transferred onto methyl cellulose/uranyl acetate mixture drops on ice for 10 min. Grids were looped out, air-dried, and analyzed by transmission electron microscopy. Electron microscopy was performed at 80 kV in an FEI Tecnai G20 microscope equipped with an SIS Veleta camera.

*Flow cytometry*. For phenotypical single EV analysis by conventional flow cytometry, the EV suspension was extensively diluted in 400 μL filtered PBS and stained for 30 min with anti-CD8-BV421 (clone RPA-T8, 1:500), anti-CD73-PE (clone AD2, 1:300), and anti-CD9-APC (clone HI9a, 1:200) antibodies as indicated. After extensive washing of the fluidics system of the cytometer, ApogeeMix (Apogee Flow Systems, silica and latex beads) and Megamix-Plus SSC (BioCytex, latex/polystyrene beads) heterogenous fluorescent beads were used to set up the system. Samples were measured at the FACSAria IIIU (BD Biosciences) for 1 min at a flow rate of 1.0 at 4 °C. Controls included buffer alone, buffer with single and combined antibodies, EVs with buffer, and EVs with fluorescence minus one (FMO) stainings for each fluorochrome. To confirm the nature of the EVs, the samples were incubated with 0.5% NP40 for 45 min and reanalyzed for 1 min (lysis control).

For the phenotypical characterization of SF EVs, we used the MACSPlex Exosome kit (Miltenyi). This bead-based assay allows the simultaneous detection of 37 surface markers to determine the cellular origin of the EVs. In brief, EVs were incubated overnight with antibody-coated capture beads, washed, and incubated with tetraspanin CD9/CD81/CD63 antibodies provided in the kit. The measurements were done at FACSCanto II (BD Biosciences).

**T cell assays**. T cells were stimulated as described above and cultured at a density of $0.25–0.5 \times 10^6$ cells/mL in serum-free X-VIVO 15 medium (Lonza) for 3 to 4 days at 37 °C, 5% $CO_2$ in the presence of ADA inhibitor EHNA (10 μM, Tocris). In Treg suppression assays, responder T cells were cocultured with Tregs at different ratios. When indicated, AMP (50 μM, Sigma-Aldrich), ATP (50 μM, Sigma-Aldrich), the CD73-specific inhibitor 2-chloro-$N^6$-o-chlorobenzyl-α,β-methylene-ADP (PSB-14685, 10 μM)[59], recombinant CD73 (15 ng/mL, unless otherwise noted, Sino Biological) or EVs (equivalent to 150 μL of cell culture supernatant of activated CD8 T cells, if not stated otherwise) were added. At the time of harvest, the cell membrane expression of the activation marker CD25, and the dilution of

eFluor 670 as a measure of proliferation were assessed by flow cytometry. IFNγ was determined in cell culture supernatants by ELISA (BioLegend), measured with a Victor[3] 1240 plate reader equipped with Wallac 1240 Manager software (PerkinElmer).

**Phospholipase C treatment**. Forced shedding of CD73 was achieved by incubating $0.2 \times 10^6$ CD8 T cells with 0.5 U/mL bacterial phosphatidylinositol-specific phospholipase C (PI-PLC, Thermo Fisher Scientific) for 30 min at 37 °C. Cells and supernatants after PI-PLC treatment were subsequently used for the determination of AMPase activity by high-performance liquid chromatography (HPLC).

**Analysis of ATPase and AMPase activities by HPLC**. AMPase and ATPase activity of cells, cell culture supernatants, EVs, and SF were analyzed by assessing the degradation of $1,N^6$-etheno-AMP (eAMP) or $1,N^6$-etheno-ATP (eATP), respectively. For this, $0.2 \times 10^6$ T cells, 150 μL cell culture supernatant, EVs (equivalent to 150 μL of supernatant), or 20 μL SF were incubated with 1 μM eAMP or eATP (Biolog) for 30 min (cells) or 60 min (cell culture supernatants, EVs and SF) at 37 °C. After the incubation, cells were removed by centrifugation ($450 \times g$, 5 min, 4 °C) and all samples were directly frozen and stored at −20 °C until HPLC analysis. The analysis of etheno-nucleotides was conducted by ion pair reversed-phase HPLC on a 1260 Infinity system (Agilent Technologies). Before measurement, samples were passed through 10 kDa size exclusion filters to remove proteins (10 min, $10,000 \times g$, 4 °C, Sartorius). A volume corresponding to starting amount of 85 pmol eAMP or eATP was loaded on the HPLC. For quantification of nucleotides in the sample, different amounts of commercially available etheno-nucleotides (Biolog) were analyzed under the same conditions. The separation was performed on a 250 mm × 4.6 mm C-18 BDS Multohyp 5 μm column (CS Chromatographie Service) with a C-18 security guard cartridge (Phenomenex). The mobile phase was composed of HPLC buffer A (20 mM $KH_2PO_4$, 5 mM TBAHP, pH 6.0) and HPLC buffer B (50% buffer A and 50% methanol) with the following gradient: 0.0 min (30.0% buffer B), 3.5 min (30.0% buffer B), 11.0 min (62.5% buffer B), 15.0 min (62.5% buffer B), 25.0 min (100.0% buffer B), 27.0 min (100.0% buffer B), 29.0 min (30.0% buffer B), and 38.0 min (30.0% buffer B). The injection volume was 100 μL and the flow rate was 0.8 mL/min. The temperature of the column compartment was 20 °C and the autosampler was kept at 8 °C. The signals were detected by the fluorescence detector (excitation 230 nm and emission 410 nm) of the system. Peak integration was performed using ChemStation Software (Agilent Technologies).

**Detection of non-cell-bound CD73**. Non-cell-bound CD73 was determined in cell culture supernatants and in SF by ELISA (Abcam).

**Fluorescence microscopy**. Human CD73[+] and CD73[−] CD8 T cells or total CD8 T cells were stimulated as described above. At the day of harvest, cells were plated on poly-L-lysine coated cytoslides (Thermo Fisher Scientific) at a density of $0.2 \times 10^6$ cells/cytoslide using a cytospin centrifuge (Shandon Elliott). In some experiments, exocytosis was blocked with 10 μM GW4869 (Cayman Chemical). Cells were fixed for 10 min at room temperature (RT) with 4% PFA (EMSciences) and washed with PBS. For immunofluorescent localization of CD73, unspecific binding was blocked with 5% normal horse serum (Vector) diluted in 0.05% Triton X-100 (Merck) in PBS. The cytoslides were incubated overnight with unconjugated anti-CD73 (clone AD2, 1:10, Biolegend) in blocking buffer at 4 °C and detected using a Cy2 anti-mouse antibody (1:200, Jackson ImmunoResearch Laboratories) for 30 min at RT. After washing, cells were further incubated with APC anti-CD9 (clone HI9A, 1:10) or/and anti-CD81 (clone 5A6, 1:10) (both BioLegend) for 60 min at RT. An LSM800 confocal microscope with airyscan and the ZENblue software (all ZEISS) were used for analysis. Pearson's coefficient of co-localization was determined in seven to ten randomly chosen high power fields (630X) per condition (three individual cells/high power field).

**Statistical analysis**. Prism 8 (GraphPad) was used to perform statistical analyses, detailed information are provided in the figure legends. In brief, data were analyzed for normal distribution. When they passed the normality test, they were analyzed by two-tailed Student's t-test (two groups, paired), ordinary one-way ANOVA (multiple groups, unpaired), or repeated measures (RM) one-way ANOVA (multiple groups, paired). When data did not pass the normality test, they were analyzed by a two-tailed Wilcoxon test (two groups, paired) or Kruskal–Wallis test (multiple groups, unpaired). When multiple groups were compared, post hoc tests were performed to correct for multiple comparisons. Dunnett's multiple comparisons test was used when all groups were compared to the same control condition. Dunn's multiple comparisons test was performed to compare the mean ranks of not normally distributed data. $p$ values $< 0.05$ were considered to indicate statistical significance. Non-significant differences were not annotated.

**Reporting Summary**. Further information on research design is available in the Nature Research Reporting Summary linked to this article.

## Data availability
The authors declare that the data supporting the findings of this study are available within the paper and its supplementary information files, or in the source data file. Data from publicly available sources shown in this paper can be obtained from the Human Protein Atlas (https://www.proteinatlas.org/ENSG00000135318-NT5E/blood, NT5E expression in human blood). Source data are provided with this paper.

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

## Acknowledgements

We thank the following investigators and technical personnel for contributing patients and samples, and for technical assistance: Romy Hackbusch, Manuela Kolster, Sandra Lipovac, Kati Tillack, Nikolay Tzaribachev, Elisabeth Weißbarth-Riedel, and the UKE FACS Core Facility. We also thank patients and blood donors, and the Department of Transfusion Medicine at the UKE for their cooperation. This work was supported by the German Research Council (SFB1328/ID: 335447717 to E.T., N.G., T.M., R.F., and C.E.M.; RI 2952/1-1 to A.R.; RI 2616/3-1 to F.L.R.; SFB1129/ID: 240245660 to C.M.-S.; FOR2879 TO235/11-1 to E.T.), the Hamburg State Excellence Research Program, the Werner Otto Foundation, the UKE intramural programs FFM (to A.R.) and (to E.S.), and the University of Hamburg (stipend to E.S.).

## Author contributions

Idea and design of research project: E.T., E.S., R.W., and N.G. Writing manuscript: R.W., E.S., and E.T. Supply of patient material: I.R. and M.L. Establishment of methods: E.S., R.W., A.R., J.B., A.B., R.F., B.P., B.R., and F.L.R. Experimental work: E.S., R.W., A.R., C.M.-S., R.R., J.B., S.B., and H.W. Data analysis and interpretation: E.S., R.W., A.R., and E.T. Scientific input and manuscript revision: A.R., C.M.-S., F.L.R., J.B., B.R., B.P., F.C., T.M., R.F., C.E.M., N.G.

## Funding

## Competing interests

C.E.M. has given scientific advice to Arcus Biosciences (Arcus Biosciences, Inc. is a publicly-traded biotechnology company working on CD73 inhibitor development). The remaining authors declare no competing interests.
