## [Peer Review File · Nature Communications]

CD73-mediated adenosine production by CD8 T cell-derived extracellular vesicles constitutes an intrinsic mechanism of immune suppressionREVIEWER COMMENTS

Reviewer #1 (Remarks to the Author):

The manuscript submitted by Schneider et al entitled 'CD73-mediated adenosine production by T cell-derived extracellular vesicles constitutes an intrinsic mechanism of immune suppression', report the differential expression of CD73 and CD39 on human Treg and Teffs (CD4 and CD8), showing that the expression of these two ectonucleotides differs between T cell subsets. They also showed that CD73 and CD39 were functional on human T cell subsets that express them using a variety of inhibitors. The authors also showed that enzymatic active CD73 is released by activated CD8+ T cells, with CD73 being associated with extracellular vesicles. These vesicles were shown to be suppressive in the presence of AMP and in the presence of Tregs they inhibited CD73-CD4+ con cells. The expression of CD73 on suppressive T cell derived EVs is not new and has been shown in a murine setting (Smyth et al 2013) with reference to regulatory T cells, something that has been omitted from the paper by the authors. Additionally, despite the lack of CD73 human Tregs derived EVs have been shown to be suppressive, something also omitted from this paper.

Although the findings of the manuscript are interesting there are a few key points that should be addressed to strengthen their findings.

Major points:

1) Given the findings of the paper I am surprised that a full description and analysis of the EVs as highlighted by the ISEV guidelines is missing. This needs to be included so that the reader can be assured of the EVs used. This should be extended to the clinical samples. In addition, and importantly, no mention of contaminants in the EV prep that could have attributed to your data are included in the paper. Again, this is important as contaminating proteins/lipids ect can be seen in ultracentrifuged EV pellets.

2) The lack of CD73, but CD39 expression, on human Treg derived EVs has been shown by Tung et al (2020), which the authors have failed to describe in their paper. Given this, have the authors undertaken their experiments described in Fig 6 in the absence of Tregs, but in the presence of CD39 expressing Treg vesicles?

3) The EVs expressing CD73 used in these experiments came from CD8+ T cells. Did the authors investigate whether vesicles expressing CD73 were isolated from CD4+ T cells? In their discussion they say CD73 on conventional T cell-derived EVs, but what they have shown is just CD8 T cell derived EVs. This should be corrected to reflect what they show in the manuscript. If this mechanism of suppression of CD4+ T cells happens in vivo, which has not been validated here, what is the likelihood that CD8+ T cell vesicles are found at the site of CD4+ T cell responses? Can the authors be sure that Tregs would not inhibit the production of these CD8 T cell vesicles under physiological conditions?

4) Which cell(s) do the CD73+EVs isolated from RA SF come from, given that the expression of CD73+ is reduced in all T subsets? They mention in their discussion that it was not B cell or monocyte derived but they have not directly shown that it is attributed to T cells so the discussion should be toned down to reflect this. Does the authors believe that this is because the CD73 is being secreted in vesicles? Does the sCD73 ELISA really show this given the variability between donors. Why did the authors use 3.1×10^8 particles? Did they titrate their EVs? Did they look at the CD73 associated with vesicles directly rather than via ELISA?

5) In Figure 1G: what is the CD25 expression on unstimulated Teffs? Is it affected by the addition of ATP and rec CD73? Tregs in this figure appear non suppressive in your cultures with CD73-CD4con cells. Are your Tregs suppressive in the presence of CD73+ CD4con or unseparated cells.

6) No gating strategies have been shown, nor the purity of sorted CD73-CD4+ cells. Sorry if I missed the later.

7) Statistics are missing on Fig 2. Data was representative for one donor in Fig 2 can the authors show the pooled data as they said that they were repeated 3 times.

Reviewer #2 (Remarks to the Author):

In this manuscript, the authors address the potential for multiple sources and function of CD73 in human cells – the latter being an important consideration. They provide compelling evidence for a contribution from activated T cells. Of note, I found the figure legends unusually complete and informative which is a welcome style. The role of extracellular vesicles in providing this enzymatic activity into the immediate environment is very novel. However, their rationale and discussion overlook the collaboration in ATP metabolism provided by all cells in the vicinity even though their data support the notion.

The notion that human Treg do not express CD73 is not universally supported by the literature (as noted in a paper they cited. Alam et al, J Inf Dis, 2009). Maybe it depends on the source of Treg or the assay? They cite a gene expression study which would be biased by the source of the cells, isolation techniques, mRNA stability and the assay used. In fact, data in supplemental Fig 1 suggest some donors had high levels. Different than mice I can accept. "None" is a bit misleading as those donors that do have higher levels may contribute more than the selected samples they measured. Perhaps some discussion of the metadata for the ImmGen data could be included in supplemental figures (as well as for their clinical subjects). The extension to a relevant model of human disease is a great strength.

I would disagree that the role of CD73 in immune suppression in humans is controversial as it is a target for checkpoint inhibition in cancer in order to boost immune responses (MED19447, BMS-986179, NZV930 and CPI-006).

Some of the technical details are superficially described. For example, in performing multiparameter flow-based assays it is important to control for nonspecific binding and compensation but they don't mention how this was done. However, their description and application of statistical analyses were amongst the best I have ever seen.

In Fig 1E, a point that has some disagreement in the literature, have the data been validated by RT PCR? Or another biologically independent assay? Have surface CD73 been shed as vesicles as Treg represented previously activated cells?

I think their data in fig 2 support the notion that at least some of the inhibition is attributable to CD73 on Treg but clearly other sources of CD73, possibly including vesicles from Treg and other cells, are in play. The title for the figure legend and tone of the text may need to be adjusted to reflect these possibilities. Similarly, Supplemental Figure 2 does not rule out a role, of note, 14685 was not used to attempt to block the inhibitory effect of Treg (but I may have missed some mention of it in a "one-liner" in the text).

Figures 3 to 7 provide the evidence that non Treg contribute significantly to the extracellular pool of adenosine and its anti-inflammatory effects. These studies are well described and seem to be well done.

Minor points

CD39 and CD73 are expressed by cells other than Treg so the release of ATP can lead to a pool of adenosine derived from many cells. There are multiple studies in mice illustrating the collaboration between different cell lineages based on their ability to generate adenosine. The inference that Treg are sole responsible, e.g. in the abstract, should be reworded throughout to reflect the broader expression of these ectoenzymes.

The anti-inflammatory effects of adenosine reach beyond T cells and include myeloid cells. Perhaps that could be discussed in more detail.

With the extrusion of extracellular vesicles, many believe that this is balanced by the internalization of other vesicles rather than just synthesis. Again, perhaps a bit of discussion on the turnover of membranes would round out the presentation.

CD73-mediated adenosine production by CD8 T cell-derived extracellular vesicles constitutes an intrinsic mechanism of immune suppression

Point-by-point response to the reviewers' comments on the manuscript

Response to Reviewer #1

The manuscript submitted by Schneider et al entitled 'CD73-mediated adenosine production by T cell-derived extracellular vesicles constitutes an intrinsic mechanism of immune suppression', report the differential expression of CD73 and CD39 on human Treg and Teffs (CD4 and CD8), showing that the expression of these two ectonucleotides differs between T cell subsets. They also showed that CD73 and CD39 were functional on human T cell subsets that express them using a variety of inhibitors. The authors also showed that enzymatic active CD73 is released by activated CD8+ T cells, with CD73 being associated with extracellular vesicles. These vesicles were shown to be suppressive in the presence of AMP and in the presence of Tregs they inhibited CD73-CD4+ con cells. The expression of CD73 on suppressive T cell derived EVs is not new and has been shown in a murine setting (Smyth et al 2013) with reference to regulatory T cells, something that has been omitted from the paper by the authors. Additionally, despite the lack of CD73 human Tregs derived EVs have been shown to be suppressive, something also omitted from this paper.

We thank the reviewer for the constructive comments, especially for making us aware of the MISEV guidelines and for requesting the analysis of the cellular origin of the synovial EVs. We believe that the answer to these points has made the manuscript substantially better. The omission of the Tung and Smyth papers was a misunderstanding from our side. We are very much aware of both, and had the Smyth paper mentioned in the first versions of the manuscript, as we did recently in a review (Schneider et al. 2019). The conclusions of the Tung et al. paper (Tung et al. 2020) are complementary to our findings: regulatory and effector cells cooperate in the metabolism of ATP to produce adenosine, either when ectonucleotidases are membrane-bound or contained in EVs.

Although the findings of the manuscript are interesting there are a few key points that should be addressed to strengthen their findings.

Major points:

1) *Given the findings of the paper I am surprised that a full description and analysis of the EVs as highlighted by the ISEV guidelines is missing. This needs to be included so that the reader can be assured of the EVs used. This should be extended to the clinical samples. In addition, and importantly, no mention of contaminants in the EV prep that could have attributed to your data are included in the paper. Again, this is important as contaminating proteins/lipids ect can be seen in ultracentrifuged EV pellets.*

We thank the reviewer for the suggestion of following the Minimal Information for Studies of Extracellular Vesicles (MISEV) guidelines (Théry et al. 2018). Accordingly, we:

- I. Analyzed different categories of EV markers by western blot: (i) tetraspanin CD81 as transmembrane protein localized in the plasma membrane, (ii) flotillin as cytosolic protein, and (iii) GM130, a protein of the Golgi apparatus, which is not present in EVs. We performed this analysis for EVs isolated from the supernatants of activated CD8 T cells (new Fig. 4c), and for EVs isolated from the synovial fluid (SF) of patients with juvenile idiopathic arthritis (JIA)

(new Supplementary Fig. 8b). All EV samples contain CD81 and flotillin. The absence of GM130 further confirms the EV nature of the particles we isolated. Material and methods (page 19) and results (pages 9 and 12) have been adapted accordingly.

New Fig. 4c | Characterization of CD8-derived extracellular vesicles by western blot. Western blot analysis of CD73 and EV markers on EVs derived from cell culture supernatants of stimulated CD73⁺ CD8 or CD73⁻ CD8 T cells. Cell lysate from stimulated CD8 T cells was used as positive control. Data are shown for three donors.

New Supplementary Fig. 8b | Characterization of synovial fluid-derived extracellular vesicles by western blot. Western blot analysis of CD73 and EV markers on EVs derived from the SF from patients with JIA. Cell lysate from stimulated CD8 T cells was used as positive control. Data are shown for three donors.

- II. Performed electron microscopy for confirmation of the vesicle characteristics and gold labeling (anti-CD73) of CD8-derived EVs (new Fig. 4e). The pictures of vesicles show the typical cup-shaped morphology with sizes between 50 to 200 nm. We could detect CD73 in the EVs derived from CD73⁺ CD8 T cells, but not from CD73⁻ CD8 T cells (new Fig. 4e). The legends and the text have been accordingly modified (page 19 for methods and page 9-10 for results).

New Fig. 4e | Electron microscopy characterization of extracellular vesicles derived from CD73⁻ (left) and CD73⁺ (right) CD8 T cells. EVs were isolated from cell culture supernatants of stimulated CD73⁻ or CD73⁺ CD8 T cells. CD73 is detected by immunogold labeling of the EVs.

We also performed electron microscopy with SF-derived EVs and added the image to the new Supplementary Fig. 8a. We have adapted the text in the results (page 12).

New Supplementary Fig. 8a | Characterization of synovial fluid-derived extracellular vesicles by electron microscopy. Representative electron microscopy image of EVs isolated from the SF of a patient with JIA.

Finally, we added MISEV-recommended characterization of EVs to the methods section (page 19). The amount of EVs isolated from our samples did not allow for a quantification of lipids, because we prioritized the phenotypical characterization of the cellular origin (see answer to point 4, new Supplementary Fig. 8c, Response letter Fig. 4). We are aware of this limitation, and we added a comment to this point in the text (page 12).

In short, we believe that the new western blots, electron microscopy images and bead-based flow cytometric analysis of SF EVs, together with the nanoparticle tracking analysis (NTA) already shown in the manuscript, strongly support the nature and quality of the EVs.

2) The lack of CD73, but CD39 expression, on human Treg derived EVs has been shown by Tung et al (2020), which the authors have failed to describe in their paper. Given this, have the authors undertaken their experiments described in Fig 6 in the absence of Tregs, but in the presence of CD39 expressing Treg vesicles?

The suppressive capacity of Treg-derived EVs has been described in several publications (reviewed in Rojas et al. 2020). Concerning specifically the mechanism of adenosine production, EVs derived from murine Tregs contain CD39 and CD73 and are suppressive (Smyth et al. 2013). Human Tregs express CD39, but the expression of CD73 is rather low (Fig. 1e-f in the manuscript, new Supplementary Fig. 1). Likewise, supernatants from activated Tregs showed ATPase, but not AMPase activity (Response letter Fig. 1). The experiments reported by Tung et al. (Tung et al. 2020) demonstrate suppression by

human Treg line-derived EVs, even if they do not contain CD73. In order to reconcile the suppressive effect with the lack of CD73, they speculate that CD39-containing Treg-derived vesicles fuse with the membrane of the CD73-expressing effector cells, thus cooperating in the production of adenosine.

Our data also emphasize the combined action of regulatory and effector T cells for producing adenosine. We show that Tregs cannot (or only minimally) support immune suppression if responder cells are CD73-negative (Fig. 1g and Fig. 6b). In addition, we show that CD73 is lost from the cell membrane two to three days after activation (Fig. 3a), and that EVs isolated from the supernatants of these activated cells contain CD73-specific AMPase activity (Fig. 4a-b). Importantly, the enzymatic activity of CD73 contained in EVs is not limited to the pericellular space of the cells of origin (in this case the responder cells), but can exert its effects in a paracrine manner at the site of inflammation or tumors. Regarding the source of CD39, Tung et al. show that it is contained in Treg-derived EVs. We also show that the major ATPase activity (CD39) comes from Treg cells (Fig. 6a), and that Tregs, even in very small ratios to T effector cells, mediate immune suppression in the presence of CD73⁺ (but not CD73⁻) EVs (Fig. 6b).

Response letter Fig. 1 | Cell culture supernatants from Tregs have ATPase activity. a-b Cell culture supernatants from activated Tregs (α CD3/ α CD28, three days) were incubated with a 1,*N*⁶-etheno-ATP (eATP) or b 1,*N*⁶-etheno-AMP (eAMP) and degradation of etheno-nucleotides was determined by HPLC, as described in the methods.

In order to determine if suppressive activity can be mediated by Treg-derived EVs, we repeated the experiments of Fig. 6b with Treg-derived EVs instead of Tregs. For the isolation of Treg-derived EVs, we sorted Tregs from PBMCs, stimulated them exactly under the same conditions as in the suppression assays (α CD3/ α CD28 for four days), and isolated the EVs from the cell culture supernatants. For a fair comparison, the amount of Treg-derived EVs used in this assay corresponds to the amount of Treg cells in Fig. 6b. The results using Treg-derived EVs are similar to the results shown in Fig. 6b: addition of Treg-derived EVs alone did not lead to a suppression of the CD73-negative responder T cells, but there was suppression with the addition of exogenous CD73 (Response letter Fig. 2). Therefore, we conclude that Tregs contribute to adenosine-mediated suppression with CD39, either membrane-bound or in form of EVs, as suggested by Tung et al. CD73, in contrast, is mostly provided by other cells, either cell-bound or in form of EVs (data from this manuscript, and Tung et al. 2020). The source of EVs include effector CD8 T cells upon activation (shown in this manuscript), B cells or tumor cells in the tumor environment (Zhang et al. 2019; Clayton et al. 2011), or stromal cells at sites of inflammation (Kerkelä et al. 2016). A paragraph to this topic is now included in the discussion (page 15). Because the topic of this work is not the EVs derived from Tregs, we have not included the figure below in the main manuscript.

Response letter Fig. 2 | The combination of Treg-derived extracellular vesicles and a non-Treg source of CD73 (recombinant CD73 or EVs from CD73⁺ CD8 T cells) results in T cell suppression. CD73⁻ CD4con T cells (100,000 cells) were stimulated with α CD3/ α CD28 in the presence of the ADA inhibitor EHNA (10 μ M) and incubated with Treg-derived EVs, ATP (50 μ M), rec. CD73 (15 ng/mL) and EVs from cell culture supernatants of sorted and activated CD8 T cells as indicated. The amount of Treg-derived EVs corresponds to the amount produced by 12,500, 25,000 and 50,000 Tregs, respectively, i.e. 1.0×10^7 EVs per 12,500 Treg cells for donor 1 and 4.8×10^6 EVs per 12,500 Treg cells for donor 2. CD25 expression and proliferation were measured after four days by flow cytometry. Data are shown for two independent Treg-derived EV samples (mean \pm SD of technical duplicates).

3) The EVs expressing CD73 used in these experiments came from CD8⁺ T cells. Did the authors investigate whether vesicles expressing CD73 were isolated from CD4⁺ T cells? In their discussion they say CD73 on conventional T cell-derived EVs, but what they have shown is just CD8 T cell derived EVs. This should be corrected to reflect what they show in the manuscript.

In our assays we used CD8 cells as 'CD73 donors' because the expression of CD73 in peripheral CD8 T cells is high (Fig. 1f). If CD4 cells express CD73, addition of AMP also results in a significant reduction of their activation and proliferation, as we have shown in Fig. 2b. This effect is CD73-specific, because it can be reverted by addition of the CD73-specific inhibitor PSB-14685.

Even though the expression of CD73 on peripheral blood CD4 T cells is lower than in CD8 T cells, they also lose CD73 expression after activation (Fig. 3a), suggesting a parallel mechanism to CD8 T cells in the production of CD73⁺ EVs. In Response letter Fig. 3, we show the comparison of CD73 expression on sorted CD8 and CD4con T cells, and the AMPase activity of the cell culture supernatants after activation. The higher frequency of CD73⁺ cells in the CD8 compartment is reflected by higher AMPase activity in the supernatants of the activated T cells. The low AMPase activity of CD4 T cell culture supernatants was not enough to turn into detectable activity in the EVs of the two donors that we tested, but because the model that we use in our experiments are CD8 T cells, we did not go further into upscale production of CD4-derived EVs.

We agree that the term ‘conventional T cell-derived EVs’ does not faithfully reflect what we show in our results. Therefore, we have specified ‘CD8 T cells’ in the title and throughout the manuscript, and adapted the final conclusions to CD8 T cells.

Response letter Fig. 3 | CD73 expression and enzymatic activity of human T cells. CD73 expression on human peripheral blood T cells was analyzed by flow cytometry. After stimulation (α CD3/ α CD28, three to four days), cell culture supernatants were harvested and incubated with 1, N^6 -etheno-AMP (eAMP), and degradation to 1, N^6 -etheno-ADO (eADO) was determined by HPLC (n = 4, mean \pm SD).

If this mechanism of suppression of CD4+ T cells happens in vivo, which has not been validated here, what is the likelihood that CD8+ T cell vesicles are found at the site of CD4+ T cell responses? Can the authors be sure that Tregs would not inhibit the production of these CD8 T cell vesicles under physiological conditions?

At sites of inflammation we mostly find different immune cell subsets, including local resident cells and circulating cells attracted by the signals provided *in situ*. We do not claim that CD73-containing vesicles derive exclusively from CD8 T cells. We show here that EVs derived from peripheral CD73⁺ CD8 cells (as a model for T cells expressing CD73) can provide the necessary AMPase activity for immune suppression. We believe that at the site of inflammation there will be different cell types contributing to CD73⁺ EVs (this point has been reinforced in the discussion of the revised manuscript, page 15). As shown below (see response to comment 4), the EVs purified from the synovial fluid of patients with JIA derive from a variety of cell types that could potentially contain CD73, among them CD4 and CD8 T cells, endothelial cells and synovial mesenchymal stem cells.

The inhibition of EV production by Tregs is an interesting aspect. We did not see a clear possibility to address this question experimentally because it is technically difficult to quantitatively assess phenotypically different vesicles in a mixture. In agreement with existing literature (Oba et al. 2019; van der Vlist et al. 2012), we observed that cell culture supernatants from stimulated CD8 T cells contained more vesicles than supernatants from unstimulated cells. We assume that mechanisms that reduce T cell activation, including different mechanisms of Treg-mediated suppression, lead to a reduced generation of CD8-derived EVs.

4) Which cell(s) do the CD73+EVs isolated from RA SF coming from, given that the expression of CD73+ is reduced in all T subsets? They mention in their discussion that it was not B cell or monocyte derived but they have not directly shown that it is attributed to T cells so the discussion should be toned down to reflect this.

This is a very interesting point, and we thank the reviewer for inspiring us to do this analysis. We were first interested in the contribution of T cells to the synovial fluid EVs. To answer this question, we used a flow cytometry-based method where EVs will bind to antibodies against surface markers immobilized on double fluorescence (FITC and PE)-coded beads (new Supplementary Fig. 8c). The detection occurs through APC-labeled anti-CD9/CD63/CD81 antibodies that recognize the tetraspanins on the EVs bound to beads (MACSPlex Exosome Kit, Miltenyi). This setup has several advantages, namely the possibility of simultaneously detecting over 30 surface markers in a small amount of sample, and an easy detection, because the readout is on beads, not on 150 nm EVs. Using this system, we were able to analyze EVs purified from the SF of four donors with juvenile idiopathic arthritis (JIA). The results are very clear, because the four samples are quite homogeneous regarding the main cellular markers, and can be summarized as follows: We found a signal for immune cell markers such as CD45, CD8, CD4, CD14, CD29, CD69, CD2, etc., as well as a strong signal for CD105, indicating the presence of endothelial cells and/or synovial mesenchymal stem cell (sMSC)-derived EVs. Among the immune cells, CD8 T cells and monocyte/macrophages gave the stronger signals. Additional T cell markers CD4 and CD2 gave a clear signal, indicating the presence of T cell-derived EVs. The CD3 antibody did not perform well in this kit, because we also did not obtain a positive signal with T cell-derived EVs used as internal control (not shown). We did not detect a signal for CD19, CD20 and CD56, suggesting that B- and NK-derived EVs are much less frequent. Monocyte/macrophage markers were prominent in these EVs; however, these cells in humans do not express CD73. Synovial MSC and endothelial cells express CD73, thus they very likely contribute to the pool of CD73-containing EVs in the SF. HLA molecules, CD44, CD24 are among the highest signals, but they are not specific to a single cell type. This assay clearly demonstrates that there are T cell-derived EVs in the SF.

New Supplementary Fig. 8c | Extracellular vesicles in the synovial fluid derive from a variety of cell types. **c** SF-derived EVs were captured by antibody-coated beads (dot plot shows the discrimination of bead populations by different fluorescence intensities, bead antigens are listed on the right side of the heat map), and detected with APC-labeled antibodies against tetraspanins CD9/CD63/CD81 by flow cytometry. The heat map shows background-corrected median fluorescence intensity of APC signals for the different bead populations for SF-derived EVs from four donors. DC: dendritic cells, EC: endothelial cells, sMSC: synovial mesenchymal stem cells.

The MACSPlex Exosome Kit does not permit co-expression analysis at single EV level. To find out if CD8 EVs have CD73 on their surface, we co-stained the vesicles with CD73 and CD8 and analyzed them using conventional flow cytometry (Response letter Fig. 4). With all necessary controls for this experiment, we were limited by the amount of material available and decided to concentrate on CD8 T cells. As shown in the Response letter Fig. 4d, some of the EVs in the SF co-express CD73 and CD8, demonstrating that CD73-containing EVs derived from CD8 cells are found in the SF of patients with

JIA (we also included this panel in the new Supplementary Fig. 8d). We did not test for the co-expression of stromal markers due to the lack of enough material for flow cytometry, but it is very likely that stromal cells contribute to CD73-positive EVs in this compartment.

The methods (page 20), results (page 12), and discussion (page 15) on the cellular origin of the SF EVs been added to the manuscript.

Response letter Fig. 4 | Characterization of synovial fluid-derived EVs by conventional flow cytometry. **a** Dot plots of calibrating Apogee and Megamix-Plus SSC beads shown as a size distribution reference. **b** Gating strategy: a first gate in the FSC-A vs. FSC-H dot plot was used to eliminate background. The next gate is the particle gate, where the event counts are annotated, and the final gate (PE-H vs. BV421-H) was used to exclude artifacts (antibody aggregates). **c** Histograms corresponding to the events in the particle gate show fluorescence staining for the sample stained with CD8 and CD73 (red), stained sample after lysis control (0.5% NP40 for 45 min, light grey), unstained EVs (blue) and buffer with antibodies (brown). **d** Co-expression of CD8 and CD73 on SF-derived EVs. The percentage of objects in the particle gate positive for analyzed antigens (gated based on FMO controls, left panel) is shown.

Does the authors believe that this is because the CD73 is being secreted in vesicles? Does the sCD73 ELISA really show this given the variability between donors.

Based on the following observations - (i) activated CD8 T cells lose CD73 from the cell surface upon activation (Fig. 3a), (ii) CD8 T cells are abundant in the SF from patients with JIA (Fig. 7a), (iii) CD8 T cells infiltrating the SF have low expression of CD73 (Fig. 7b), and (iv) some EVs in the SF co-express CD8 and CD73 (new Supplementary Fig. 8d) - we conclude that T cell activation in the inflamed joints leads to release of CD73 from the cell membrane, at least of CD8 T cells, and to the generation of CD73⁺ EVs.

SF is aspirated from the inflamed joints of the patients as part of therapy. The amount of CD73, soluble or EV-bound, in the SF may vary according to disease activity, treatment, infiltrating cells, etc. To establish a correlation between the concentration of CD73, the numbers of vesicles and enzymatic

activity it would be necessary to study a large cohort of patients where all variables could be considered.

We measured a reduction of SF-mediated AMPase activity in the SF after differential centrifugation steps and in SF-derived EVs (Response letter Fig. 5). The AMPase activity of the SF-derived EVs is CD73-specific, but lower than that of the SF (Fig. 7d). Because the sCD73 ELISA does not distinguish between soluble and EV-bound CD73, we conclude that CD73 is probably present in the SF both as soluble protein and bound to EVs. We show in Fig. 7e that purified SF-derived EVs are suppressive in a CD73-dependent manner (Fig. 7e); however, we cannot exclude that a soluble form of CD73 also contributes to adenosine generation *in vivo*.

Response letter Fig. 5 | Extracellular vesicles from synovial fluid of patients with juvenile idiopathic arthritis have AMPase activity. **a** Synovial fluid after different steps of differential centrifugation or **b** SF-derived EVs were incubated with 1,*N*⁶-etheno-AMP (eAMP) and conversion to 1,*N*⁶-etheno-adenosine (eADO) was determined by HPLC (mean ± SD). PSB-14685 (10 μM) was used to block CD73-mediated AMPase activity.

Why did the authors use 3.1x10⁸ particles? Did they titrate their EVs? Did they look at the CD73 associated with vesicles directly rather than vis ELISA?

We titrated EVs both from activated CD8 T cell supernatants and from synovial fluid in a suppression assay.

For the synovial fluid, we decided the number of particles to test in T cell assays based on the AMPase activity detected in 20 μL of SF (measured by HPLC). This volume corresponded to 3.1×10^8 particles, and this amount gave us a clear result in the suppression assay. To see a dose-dependent effect, we titrated down the particles until no suppressive activity was observed (new Supplementary Fig. 8e). These values (volume and number of EVs) were used as a reference for subsequent SF donors, which were then titrated within a similar range. Not surprisingly, we found interindividual variability in the suppressive activity, as shown for the two donors in the figure. In samples with limited amount of material we relied on these estimations for the number of EVs in the suppression assay. The titration of the SF-derived EVs is shown in the new Supplementary Fig. 8e.

New Supplementary Fig. 8e | Extracellular vesicles derived from synovial fluid are suppressive in a dose-dependent manner. CD73⁻ CD4con T cells were stimulated with α CD3/ α CD28 in the presence of the ADA inhibitor EHNA (10 μ M) and incubated with AMP (50 μ M) and EVs derived from SF as indicated. CD25 expression and proliferation were measured after four days by flow cytometry. Data are shown for two donors from independent experiments (mean \pm SD of technical duplicates). 3.1×10^8 particles (used in Fig. 7e) correspond to 3.5 μ L (donor 1) or 1.3 μ L (donor 2) in the graph.

We also performed a titration of the EVs derived from CD8 T cells in a similar experiment. In this case, the starting point was the amount of EVs contained in 150 μ L supernatant of 1×10^6 seeded cells/mL medium. This corresponded to 1 - 2 μ L of the EV sample, depending on the concentration factor after the isolation procedure, and this amount was titrated down until no suppression activity was observed. Again, we could see that CD8-derived EVs suppressed the responder T cells in a dose-dependent manner. We included these results in the manuscript as new Supplementary Fig. 7.

New Supplementary Fig. 1 | Extracellular vesicles derived from CD73⁺ CD8 T cells decrease the activation and proliferation of responder T cells in a dose-dependent manner. CD73⁻ CD4con T cells were stimulated with α CD3/ α CD28 in the presence of the ADA inhibitor EHNA (10 μ M) and incubated with AMP (50 μ M) and EVs derived from CD73⁺ CD8 or CD73⁻ CD8 T cells as indicated. CD25 expression and proliferation were measured after four days by flow cytometry. Data are shown for two donors from independent experiments (mean \pm SD of technical duplicates).

5) In Figure 1G: what is the CD25 expression on unstimulated Teffs? Is it affected by the addition of ATP and rec CD73?

The frequency of cells expressing CD25 on unstimulated responder cells (Fig. 1g) after four days in culture is 8.6% (compared to 39% CD25⁺ on stimulated cells), and is not altered by addition of ATP, rec. CD73 or the combination of both (Response letter Fig. 6)

Response letter Fig. 6 | CD25 expression on unstimulated CD73⁻ CD4con T cells is not affected by the addition of ATP or recombinant CD73. **a** CD25 expression on unstimulated and stimulated (α CD3/ α CD28 for four days), EHNA-treated CD73⁻ CD4con T cells. **b** Unstimulated CD73⁻ CD4con T cells were treated with the ADA inhibitor EHNA (10 μ M) and incubated with ATP (50 μ M) and recombinant CD73 (15 ng/mL) as indicated. CD25 expression was measured after four days by flow cytometry.

Tregs in this figure appear non suppressive in your cultures with CD73-CD4con cells. Are your Tregs suppressive in the presence of CD73+ CD4con or unseparated cells.

In Fig. 1g we aimed to elucidate the relevance of Tregs as producers of adenosine (having both ATPase and AMPase activities) in the human system. To address this point, we had to completely strip the system from AMPase activity. Therefore, we used CD73-negative responder cells and serum-free medium. Under these conditions, Tregs appear less suppressive than what we are used to see in an assay with serum and responder cells expressing CD73 (i.e. PBMCs). The poor immune suppressive capacity in our setup may be due to the absence of APCs that could provide additional co-stimulation (other than α CD28) to the Tregs, or to the lack of growth factors/hormones normally present in the serum. Our Tregs are indeed suppressive in a classical *in vitro* suppression assay: In the figures below, we show the gating strategy (Response letter Fig. 7) and the results of the suppression assay using (a) PBMCs or (b) conventional CD4 cells as responders (Response letter Fig. 8)

Response letter Fig. 7 | Gating strategy to determine the activation and proliferation of responder T cells in a suppression assay containing regulatory T cells. After gating on lymphocytes, doublets and dead cells were excluded from the analysis. Expression of CD25 and eFluor 670 dilution on CD4⁺ and CD8⁺ cells were determined as markers of activation and proliferation, respectively. The example shown corresponds to PBMCs as responder cells seeded at a ratio of 100,000 PBMCs to 50,000 Tregs (1 : 0.5) three days after stimulation.

Response letter Fig. 8 | Regulatory T cells suppress the proliferation of CD4 T cells. **a** PBMCs were stimulated with α CD3 and co-cultured with Tregs. **b** CD4con T cells were stimulated with α CD3/ α CD28 and co-cultured with Tregs. The ratios of responder T cells to Tregs were 1 : 0.125 to 1 : 2. Proliferation was measured three days after stimulation by flow cytometry.

6) No gating strategies have been shown, nor the purity of sorted CD73-CD4+ cells. Sorry if I missed the later.

We sort our cells with a FACSaria IIIU sorter with settings for high purity (4-way-purity sort). We usually achieved a purity of >95% for Tregs and CD4con, and went for a purity higher than 99% of CD73⁻ CD8 T cells and CD73⁻ CD4 T cells. The gate for CD73⁺ T cells was less stringent, because a high purity is not necessary for this condition, and we favored higher recovery in order to have more responder cells. The figure below shows the (a) purity of the sorted populations and the (b) gating strategy for the T cell assay and is depicted in the new Supplementary Fig. 2.

New Supplementary Fig. 2 | Purity of cell populations after cell sorting and gating strategy for T cell assays. **a** Purity of cell populations after cell sorting. **b** Gating strategy to determine activation and proliferation of CD4con T cells.

7) Statistics are missing on Fig 2. Data was representative for one donor in Fig 2 can the authors show the pooled data as they said that they were repeated 3 times.

The experiments corresponding to Fig. 2 were performed in three to nine donors. We have now included the results as a summary of all donors with statistical analysis instead of showing one representative case (new Fig. 2). We have purposely avoided the calculation of relative values and instead decided to show the raw data for each donor. The same figure in a donor color-coded version is shown below (Response letter Fig. 9), so that interindividual differences can be appreciated.

Response letter Fig. 9 | Treg-derived CD73 is not essential for adenosine-mediated immune suppression of conventional CD4 T cells. **a-d** CD4 responder T cells were stimulated with α CD3/ α CD28 in the presence of the ADA inhibitor EHNA (10 μ M). CD25 expression and proliferation were measured after four days by flow cytometry. **a** CD4con T cells were stimulated and incubated with AMP (50 μ M) and Tregs in the indicated ratio. **b** CD4con T cells were sorted into CD73⁻ and CD73⁺, the cells were incubated with AMP (50 μ M) and the specific CD73 inhibitor PSB-14685 (10 μ M). **c** CD4con CD73⁻ T cells were incubated with AMP (50 μ M), the specific CD73 inhibitor PSB-14685 (10 μ M) and recombinant CD73 (15 ng/mL). **d** CD4con CD73⁻ T cells were incubated with AMP (50 μ M) and different concentrations of soluble recombinant CD73 (three ten-fold serial dilutions starting with 15 ng/mL). PSB-14685 was used to block the highest concentration of recombinant CD73. Data are shown for three to nine donors (mean \pm SD). Ordinary one-way ANOVA with Dunnett's multiple comparisons test was used to compare all conditions to cells treated with EHNA or EHNA and AMP (first bar) (* $p < 0.05$, ** $p < 0.01$, *** $p < 0.001$).

Response to Reviewer #2

In this manuscript, the authors address the potential for multiple sources and function of CD73 in human cells – the latter being an important consideration. They provide compelling evidence for a contribution from activated T cells. Of note, I found the figure legends unusually complete and informative which is a welcome style. The role of extracellular vesicles in providing this enzymatic activity into the immediate environment is very novel. However, their rationale and discussion overlook the collaboration in ATP metabolism provided by all cells in the vicinity even though their data support the notion.

We thank the reviewer for recognizing the novelty of our work. In response to his last point we have now better addressed the point about the collaboration in ATP metabolism provided by different cells: Our assays are done in a system where the enzymatic activity for adenosine generation and degradation is controlled: responder cells are sorted with high purity to be positive or negative for CD73 (new Supplementary Fig. 2), CD73 is exogenously added (recombinant protein or EVs), and the assays are performed in serum-free medium (to avoid unpredicted enzymatic activity). Under these conditions, we can exclude the involvement of other adenosine-generating cell types or AMPase sources in our assays. It is difficult to determine if the ATPase activity in our assays comes from Tregs, activated responder cells, or both. We could show that, even under stimulated conditions, Tregs are superior to CD4con and CD8 cells in ATP degradation (Fig. 6a).

In vivo, however, the enzymatic machinery to degrade ATP and produce adenosine is present in many cell types, both immune cells and stromal cells, and they all contribute to the degradation of ATP and production of adenosine. To illustrate the complexity of EV sources in a setting of inflammation, we have traced the cellular origin of the EVs isolated from the synovial fluid of four patients with JIA and showed that both immune-derived (CD45⁺) and stromal-derived (CD105⁺) EVs are present in the SF (new Supplementary Fig. 8c). The contribution of the different cell types to ATP metabolism has now been emphasized in the discussion (page 15).

The notion that human Treg do not express CD73 is not universally supported by the literature (as noted in a paper they cited. Alam et al, J Inf Dis, 2009). Maybe it depends on the source of Treg or the assay? They cite a gene expression study which would be biased by the source of the cells, isolation techniques, mRNA stability and the assay used. In fact, data in supplemental Fig 1 suggest some donors had high levels. Different than mice I can accept. "None" is a bit misleading as those donors that do have higher levels may contribute more than the selected samples they measured. Perhaps some discussion of the metadata for the ImmGen data could be included in supplemental figures (as well as for their clinical subjects). The extension to a relevant model of human disease is a great strength.

The reviewer is right that the expression of CD73 on T cells depends on the tissue. In the intestine, for instance, as shown by Alam et al. (Alam et al. 2009) (and our own unpublished data confirm this finding), the expression of CD73 on CD4 and CD8 T cells is higher than in peripheral blood. In contrast, the expression of CD73 in T cells of the synovial fluid is lower (Botta Gordon-Smith et al. 2015 and Fig. 7b of the manuscript). It is also known that environmental cues (microbiome, diet, pathogen exposure) (Mangino et al. 2017) influence the expression of CD73 in Tregs. In addition, the fact that after activation the expression of CD73 is transiently upregulated (one to two days after activation, depending on the donor) before disappearing from the cell membrane contributes to conflicting reports on the expression of CD73 on activated T cells. A further source of discrepancy is the gating strategy, which ten years ago involved only high expression of CD25, and a far from perfect

intracellular staining of FOXP3. Moreover, the identification of human Tregs after activation is not trivial, because both CD25 and FOXP3 are upregulated in CD4 conventional cells after activation (Allan *et al.* 2007).

The Supplementary Fig. 1a shows the transcriptional profiles in matched B, NK and T cell subpopulations between human and mouse, according to data from Shay *et al.* (Shay *et al.* 2013). Each column in each of the three panels represents a subpopulation of B, NK or T cells, not individual donors or mice. This figure shows that, while *Nt5e* expression is negligible in most B cell subpopulations in mice, human B cells are strongly positive for the CD73 gene. In NK cells it is the other way around. In T cells, CD73 gene expression is very high in most murine T cell subpopulations, while in humans only few of the subpopulations are strongly positive. We have now changed the legend of this figure to make it clearer to the reader, and have added an additional panel where the expression of *NT5E* is shown for specific human immune cell populations (new Supplementary Fig. 1b, data obtained from the Human Protein Atlas (Uhlen *et al.* 2019; “Blood Atlas - NT5E - The Human Protein Atlas”). These data clearly show the low expression of *NT5E* in *ex vivo* human Tregs obtained from peripheral blood.

New Supplementary Fig. 1 | Human and murine T cell populations differ in the expression of CD39 and CD73.

a Transcriptional profiles in matched B, NK and T cell subpopulations between mouse and human (data were obtained from the ImmGen database (Heng *et al.* 2008; Shay *et al.* 2013)). **b** Gene expression profile of *NT5E* (encoding CD73) in selected human immune cell populations (data obtained from the Human Protein Atlas (Uhlen *et al.* 2019; “Blood Atlas - NT5E - The Human Protein Atlas”). **c** Representative dot plots of CD73 and CD39 expression on murine T cell subsets. Cells derived from the spleen of C57BL/6 mice.

I would disagree that the role of CD73 in immune suppression in humans is controversial as it is a target for checkpoint inhibition in cancer in order to boost immune responses (MEDI9447, BMS-986179, NZV930 and CPI-006).

We have realized that the following sentence in our introduction

‘While several studies demonstrate the importance of CD39 expression on human Tregs for their suppressive capacity (Borsellino *et al.* 2007; Fletcher *et al.* 2009; Rissiek *et al.* 2015), the evidence for a role of CD73 is controversial (Mohammad S. Alam *et al.* 2009; Gourdin *et al.* 2018).’

is indeed an unfortunate sentence. We apologize for the mistake. We do not question in any way the role of CD73 in immune suppression in humans. We have plenty of data to support it. The sentence on page 4/5, lines 86-88 has now been changed to

‘[...] the evidence for an **essential** role of CD73 on Tregs is controversial (Alam *et al.* 2009; Gourdin *et al.* 2018; Tung *et al.* 2020).’

Some of the technical details are superficially described. For example, in performing multiparameter flow-based assays it is important to control for nonspecific binding and compensation but they don't mention how this was done. However, their description and application of statistical analyses were amongst the best I have ever seen.

We thank the reviewer for this comment, and have now added detail to the description of the methods. We routinely titrate all antibodies for specific purposes, and perform flow cytometry experiments with all necessary controls. The staining cocktails are designed to minimize the effects of spectral overlap. The compensation matrices are established for each antibody cocktail used, as described in a technical paper for an ‘Optimized Multicolor Immunofluorescence Panel, OMIP’ that we recently published in *Cytometry Part A* (Bremer *et al.* 2021). We have now added a new figure depicting the purity of the isolated populations and the gating strategy used for the suppression assays (new Supplementary Fig. 2), and the text to the methods in flow cytometry has been changed to add more detail (page 18).

Also the characterization of EVs has now been described in all details in the material and methods section (pages 19-20) according to the MISEV guidelines (Théry *et al.* 2018). For single EV analysis by conventional flow cytometry, we show all necessary controls in Response letter Fig. 4 and adapted the method in the manuscript accordingly (page 20).

In Fig 1E, a point that has some disagreement in the literature, have the data been validated by RT-PCR? Or another biologically independent assay? Have surface CD73 been shed as vesicles as Treg represented previously activated cells?

We agree that the complete absence of CD73 on Tregs is very rare, and the example chosen in our original Fig. 1e is not representative. We have now changed the figure using data from another donor that represents better the expression profile of CD73 on human T cells (new Fig. 1e). In the figure below (Response letter Fig. 10), the expression of CD73 on Tregs (same data as in Fig. 1f), can be appreciated better using a different scale. Indeed, the frequency of CD73-expressing Treg cells ranges between 0.8% and 6% (as stated in the results section, page 6, line 128), and of the cells co-expressing CD39 and CD73 between 0% and 3.4%.

New Fig. 1e | Representative dot plots showing the expression of CD73 and CD39 on human T cell subsets.

Response letter Fig. 10 | Detail of Fig. 1f showing the expression of CD73 on human Tregs and CD4 T cells. Data are shown for ten donors (median).

Moreover, there are plenty of data from published databases where it is shown that Tregs have the lowest *NT5E* expression among human T cells. Below we show *NT5E* expression in two independent datasets (Response letter Fig. 11). Data were obtained from the website of the Human Protein Atlas (Uhlen *et al.* 2019; “Blood Atlas - *NT5E* - The Human Protein Atlas”).

HPA scaled dataset¹

Monaco scaled dataset¹

Response letter Fig. 11 | *NT5E* gene expression in selected human peripheral blood cell types. Data obtained from the Human Protein Atlas (Uhlen *et al.* 2019; “Blood Atlas - *NT5E* - The Human Protein Atlas”). Image credit to the Human Protein Atlas.

The question if surface CD73 has been shed in Tregs in relation to previous activation is interesting. We have not addressed this here; however, data from our healthy cohort (Gehbauer *et al.*, unpublished data) show a tendency for an inverse correlation between the percentage of HLA-DR⁺ (activated) Tregs and the frequency of CD73⁺ Tregs (Response letter Fig. 12). This could mean that it is indeed the case that Tregs have lost CD73 upon previous activation. Still, the frequency of CD73 is not higher than 10% in non-activated Tregs.

Response letter Fig. 12 | CD73 expression in activated Tregs. CD73 expression on Tregs was analyzed in a cohort of 40 healthy donors (between one and 83 years of age) and correlated to the activation status of the Tregs (simple linear regression).

I think their data in fig 2 support the notion that at least some of the inhibition is attributable to CD73 on Treg but clearly other sources of CD73, possibly including vesicles from Treg and other cells, are in play. The title for the figure legend and tone of the text may need to be adjusted to reflect these possibilities. Similarly, Supplemental Figure 2 does not rule out a role, of note, 14685 was not used to attempt to block the inhibitory effect of Treg (but I may have missed some mention of it in a “one-liner” in the text).

Tregs definitively contribute to immune suppression and they do so using different mechanisms, such as sequestering of IL-2, downmodulation of APCs, production of regulatory cytokines, and production of adenosine. This is clearly shown in Response letter Fig. 8, in a classical suppression assay using PBMCs as responder cells (CD4⁺ T cells shown), and increasing amounts of Tregs. In that setting, CD73 (membrane-bound or by EVs) is provided by different cell types, namely the responder CD4⁺ T cells, and, in the left panel of Response letter Fig. 8, by CD8 and B cells also present in the assay. In the assays described in Fig. 2 we do not have additional cell types apart from responder CD4 T cells and Tregs. In this very simplified and controlled system, CD73 can only come from responder cells (unless they are specifically sorted CD73⁻) or from Tregs. It has also previously been shown that human Treg-derived EVs do not contain CD73 (Tung et al. 2020).

Fig. 2a also shows a significant decrease in T cell activation and proliferation in the presence of Tregs. Here, though, it cannot be concluded that CD73 on Tregs plays an essential role, because when AMP was added to the system to provide substrate for CD73, suppression was occurring in the absence of Tregs. In the next figures (Fig. 2b-d, and in the related Supplementary Fig. 3 showing IFN γ production), though, we clearly show that if the responder cells are CD73⁻, there is no suppression even if AMP is added, and this is reversed by addition of recombinant CD73. The point that we want to make in this very controlled system shown in Fig. 2 is that CD73 does not have to come from Tregs, but the reviewer is right that the contribution of Treg-CD73 in Fig. 2a cannot be disregarded. Therefore, we have changed the title of Fig. 2 from

‘Fig. 2 Adenosine-mediated suppression of conventional CD4 T cells is independent of Treg-derived CD73.’

to

‘Fig. 2 Treg-derived CD73 is not essential for adenosine-mediated immune suppression of conventional CD4 T cells.’

The text in the results section has been adapted accordingly (page 7).

In Fig. 2a we did not add PSB-14685 because the inhibitor would not only block CD73 on Tregs, but also on the responder cells.

Figures 3 to 7 provide the evidence that non Treg contribute significantly to the extracellular pool of adenosine and its anti-inflammatory effects. These studies are well described and seem to be well done.

Minor points

CD39 and CD73 are expressed by cells other than Treg so the release of ATP can lead to a pool of adenosine derived from many cells. There are multiple studies in mice illustrating the collaboration between different cell lineages based on their ability to generate adenosine. The inference that Treg are sole responsible, e.g. in the abstract, should be reworded throughout to reflect the broader expression of these ectoenzymes.

The reviewer is right that the sentence in the abstract could be misinterpreted. Therefore, we have removed the reference to the Tregs in this sentence (page 3). It reads now:

‘ATP is released from many cell types upon activation and stress, and the stepwise hydrolysis of extracellular ATP by ectonucleotidases CD39 and CD73 generates adenosine, a potent immune suppressor.’

In addition, we have emphasized the role of multiple cell types contributing to ATP metabolism in the discussion (page 15).

The anti-inflammatory effects of adenosine reach beyond T cells and include myeloid cells. Perhaps that could be discussed in more detail.

In page 13, lines 339-341 we state that adenosine can suppress a variety of immune cells by activating P1 receptors A_{2A} and A_{2B}. Indeed, T cells express constitutively A_{2A} receptors, and upregulate A_{2B} upon activation, while myeloid cells express A_{2B} constitutively. The affinity of adenosine to each receptor and the interaction between the receptors will determine the netto effect on the target cell. This is a highly interesting and complex topic, but for the sake of brevity we decided not to discuss it any further.

With the extrusion of extracellular vesicles, many believe that this is balanced by the internalization of other vesicles rather than just synthesis. Again, perhaps a bit of discussion on the turnover of membranes would round out the presentation.

The fate of EVs on the target cells constitutes an exciting field of research, because the fusion of the EV to the target cells may have functional consequences. The study of this topic, however, is still in its infancy and we can only speculate. We have introduced the following sentence to the discussion (page 14):

‘The generation of EVs poses the question of how the cells recover the membrane loss. A possibility would be the fusion with foreign EVs, which would permit a very dynamic exchange of membrane components between cells and acquisition of new functions (Prada and Meldolesi 2016; Joly and Hudrisier 2003).’

References

- Alam, Mohammad S., Courtney C. Kurtz, Robert M. Rowlett, Brian K. Reuter, Elizabeth Wiznerowicz, Soumita Das, Joel Linden, Sheila E. Crowe, and Peter B. Ernst. 2009. "CD73 Is Expressed by Human Regulatory T Helper Cells and Suppresses Proinflammatory Cytokine Production and Helicobacter Felis –Induced Gastritis in Mice." *The Journal of Infectious Diseases* 199 (4): 494–504. <https://doi.org/10.1086/596205>.
- Allan, Sarah E., Sarah Q. Crome, Natasha K. Crellin, Laura Passerini, Theodore S. Steiner, Rosa Bacchetta, Maria G. Roncarolo, and Megan K. Levings. 2007. "Activation-Induced FOXP3 in Human T Effector Cells Does Not Suppress Proliferation or Cytokine Production." *International Immunology* 19 (4): 345–54. <https://doi.org/10.1093/intimm/dxm014>.
- "Blood Atlas - NT5E - The Human Protein Atlas." n.d. Accessed May 7, 2021. <https://www.proteinatlas.org/ENSG00000135318-NT5E/blood>.
- Botta Gordon-Smith, Sophie, Simona Ursu, Simon Eaton, Halima Moncrieffe, and Lucy R. Wedderburn. 2015. "Correlation of Low CD73 Expression on Synovial Lymphocytes with Reduced Adenosine Generation and Higher Disease Severity in Juvenile Idiopathic Arthritis." *Arthritis and Rheumatology* 67 (2): 545–54. <https://doi.org/10.1002/art.38959>.
- Bremer, Sarah-Jolan, Laura Glau, Christina Gehbauer, Annika Boxnick, Daniel Biermann, Jörg Siegmund Sachweh, Eva Tolosa, and Anna Gieras. 2021. "OMIP 073: Analysis of Human Thymocyte Development with a 14-Color Flow Cytometry Panel." *Cytometry. Part A*, March. <https://doi.org/10.1002/cyto.a.24326>.
- Clayton, Aled, Saly Al-Taei, Jason Webber, Malcolm D. Mason, and Zsuzsanna Tabi. 2011. "Cancer Exosomes Express CD39 and CD73, Which Suppress T Cells through Adenosine Production." *The Journal of Immunology* 187 (2): 676–83. <https://doi.org/10.4049/jimmunol.1003884>.
- Gourdin, Nicolas, Marion Bossennec, Céline Rodriguez, Selena Vigano, Christelle Machon, Camilla Jandus, David Bauché, et al. 2018. "Autocrine Adenosine Regulates Tumor Polyfunctional CD73+CD4+ Effector T Cells Devoid of Immune Checkpoints." *Cancer Research* 78 (13): 3604–18. <https://doi.org/10.1158/0008-5472.CAN-17-2405>.
- Heng, Tracy S P, Michio W Painter, The Immunological Genome Project Consortium, Kutlu Elpek, Veronika Lukacs-Kornek, Nora Mauermann, Shannon J Turley, et al. 2008. "The Immunological Genome Project: Networks of Gene Expression in Immune Cells." *Nature Immunology* 9 (10): 1091–94. <https://doi.org/10.1038/ni1008-1091>.
- Joly, Etienne, and Denis Hudrisier. 2003. "What Is Trophocytosis and What Is Its Purpose?" *Nature Immunology* 4 (9): 815. <https://doi.org/10.1038/ni0903-815>.
- Kerkelä, Erja, Anita Laitinen, Jarkko Rabinä, Sami Valkonen, Maarit Takatalo, Antti Larjo, Johanna Veijola, et al. 2016. "Adenosinergic Immunosuppression by Human Mesenchymal Stromal Cells Requires Co-Operation with T Cells." *Stem Cells* 34 (3): 781–90. <https://doi.org/10.1002/stem.2280>.
- Mangino, Massimo, Mario Roederer, Margaret H. Beddall, Frank O. Nestle, and Tim D. Spector. 2017. "Innate and Adaptive Immune Traits Are Differentially Affected by Genetic and Environmental Factors." *Nature Communications* 8 (January): 13850. <https://doi.org/10.1038/ncomms13850>.
- Oba, Ryutaro, Motomichi Isomura, Akira Igarashi, and Kinuya Nagata. 2019. "Circulating CD3+HLA-DR+ Extracellular Vesicles as a Marker for Th1/Tc1-Type Immune Responses." *Journal of Immunology Research* 2019. <https://doi.org/10.1155/2019/6720819>.
- Prada, Ilaria, and Jacopo Meldolesi. 2016. "Binding and Fusion of Extracellular Vesicles to the Plasma Membrane of Their Cell Targets." *International Journal of Molecular Sciences* 17 (8): 1296. <https://doi.org/10.3390/ijms17081296>.
- Rojas, Carolina, Mauricio Campos-Mora, Ignacio Cárcamo, Natalia Villalón, Ahmed Elhousseiny, Pamina Contreras-Kallens, Aarón Refisch, et al. 2020. "T Regulatory Cells-derived Extracellular Vesicles and Their Contribution to the Generation of Immune Tolerance." *Journal of Leukocyte Biology* 108 (3): 813–24. <https://doi.org/10.1002/JLB.3MR0420-533RR>.

- Schneider, Enja, Anne Rissiek, Riekje Winzer, Berta Puig, Björn Rissiek, Friedrich Haag, Hans-Willi Mittrücker, Tim Magnus, and Eva Tolosa. 2019. "Generation and Function of Non-Cell-Bound CD73 in Inflammation." *Frontiers in Immunology* 10: 1729. <https://doi.org/10.3389/fimmu.2019.01729>.
- Shay, Tal, Vladimir Jovic, Or Zuk, Katherine Rothamel, David Puyraimond-Zemmour, Ting Feng, Ei Wakamatsu, et al. 2013. "Conservation and Divergence in the Transcriptional Programs of the Human and Mouse Immune Systems." *Proceedings of the National Academy of Sciences of the United States of America* 110 (8): 2946–51. <https://doi.org/10.1073/pnas.1222738110>.
- Smyth, Lesley Ann, Kulachelvy Ratnasothy, Julia Y.S. Tsang, Dominic Boardman, Alice Warley, Robert Lechler, and Giovanna Lombardi. 2013. "CD73 Expression on Extracellular Vesicles Derived from CD4+CD25+Foxp3+ T Cells Contributes to Their Regulatory Function." *European Journal of Immunology* 43 (9): 2430–40. <https://doi.org/10.1002/eji.201242909>.
- Théry, Clotilde, Kenneth W. Witwer, Elena Aikawa, Maria Jose Alcaraz, Johnathon D. Anderson, Ramarason Andriantsitohaina, Anna Antoniou, et al. 2018. "Minimal Information for Studies of Extracellular Vesicles 2018 (MISEV2018): A Position Statement of the International Society for Extracellular Vesicles and Update of the MISEV2014 Guidelines." *Journal of Extracellular Vesicles* 7 (1): 1535750. <https://doi.org/10.1080/20013078.2018.1535750>.
- Tung, Sim Lai, Giorgia Fanelli, Robert Ian Matthews, Jordan Bazoer, Marilena Letizia, Gema Vizcay-Barrena, Farid N. Faruqu, et al. 2020. "Regulatory T Cell Extracellular Vesicles Modify T-Effector Cell Cytokine Production and Protect Against Human Skin Allograft Damage." *Frontiers in Cell and Developmental Biology* 8 (May): 317. <https://doi.org/10.3389/fcell.2020.00317>.
- Uhlen, Mathias, Max J. Karlsson, Wen Zhong, Abdellah Tebani, Christian Pou, Jaromir Mikes, Tadepally Lakshminanth, et al. 2019. "A Genome-Wide Transcriptomic Analysis of Protein-Coding Genes in Human Blood Cells." *Science* 366 (6472): eaax9198. <https://doi.org/10.1126/science.aax9198>.
- Vlist, Els J. van der, Ger J.A. Arkesteijn, Chris H.A. van de Lest, Willem Stoorvogel, Esther N.M. Nolte t. Hoen, and Marca H.M. Wauben. 2012. "CD4+ T Cell Activation Promotes the Differential Release of Distinct Populations of Nanosized Vesicles." *Journal of Extracellular Vesicles* 1 (1): 18364. <https://doi.org/10.3402/jev.v1i0.18364>.
- Zhang, Fanghui, Rongrong Li, Yunshan Yang, Chunhui Shi, Yingying Shen, Chaojie Lu, Yinghu Chen, et al. 2019. "Specific Decrease in B-Cell-Derived Extracellular Vesicles Enhances Post-Chemotherapeutic CD8+ T Cell Responses." *Immunity* 50 (3): 738-750.e7. <https://doi.org/10.1016/j.immuni.2019.01.010>.

REVIEWER COMMENTS

Reviewer #1 (Remarks to the Author):

The rebuttal and resubmitted manuscript by Schneider et al has addressed many of my previous concerns and I thank them for their responses and the inclusion of some very interesting data in the rebuttal.

I have a few minor points that need to be addressed and which will strengthen the overall story.

1) Although the authors have included a validation of their EVs, which has strengthened their arguments, they have not included any data on contaminants such as serum albumin which has been shown previously to be a contaminant of EV pellets from UC. Can they authors please add this information as well as include evidence that 'media only' EVs (ie EVs from the media used to culture their cells) do not contribute to the results they see. Given that they have not used a sucrose gradient to isolate their EVs contamination is a problem and needs to be included to allow the reader a chance to interpret the data shown.

2) On page 221, can the authors change the EV definition to 'EV enriched pellet' for the reason above.

3) In the rebuttal the authors the authors mention go pg 4, 'they speculate that CD39-containing Treg-derived vesicles fuse with the membrane of the CD73-expressing effector cells, thus cooperating in the production of adenosine' and that 'Regarding the source of CD39, Tung et al. show that it is contained in Treg-derived EVs. We also show that the major ATPase activity (CD39) comes from Treg cells (Fig. 6a), and that Tregs, even in very small ratios to T effector cells, mediate immune suppression in the presence of CD73+ (but not CD73—) EVs (Fig. 6b)'. However this has not really been discussed in the discussion. I think that they have to modify the following statement in their discussion, 'and propose a coordinated effort involving CD39 on Tregs for the degradation of ATP to AMP, and CD73 on T cell-derived EVs that provide the necessary AMPase activity to generate adenosine', to include the fact that the Treg EVs have CD39 also and maybe contributing.

4) On pg 12 of the discussion, one reference Deaglio 2007 is not in the appropriate format.

Reviewer #2 (Remarks to the Author):

It appears the authors have responded comprehensively to both reviewers. I would agree that some questions are beyond the scope of this manuscript.

CD73-mediated adenosine production by CD8 T cell-derived extracellular vesicles constitutes an intrinsic mechanism of immune suppression

Point-by-point response to the reviewers' comments on the manuscript

Response to Reviewer #1

The rebuttal and resubmitted manuscript by Schneider *et al* has addressed many of my previous concerns and I thank them for their responses and the inclusion of some very interesting data in the rebuttal. I have a few minor points that need to be addressed and which will strengthen the overall story.

We thank the reviewer for the feedback and for further comments.

1) Although the authors have included a validation of their EVs, which has strengthened their arguments, they have not included any data on contaminants such as serum albumin which has been shown previously to be a contaminant of EV pellets from UC. Can they authors please add this information as well as include evidence that 'media only' EVs (ie EVs from the media used to culture their cells) do not contribute to the results they see. Given that they have not used a sucrose gradient to isolate their EVs contamination is a problem and needs to be included to allow the reader a chance to interpret the data shown.

We agree with the reviewer that EV pellets derived from ultracentrifugation can contain contaminants, and we have addressed this issue in the EVs derived from cell culture supernatants and of synovial fluid (SF).

Fetal calf serum (FCS) is known to be a source of EVs and even EV-depleted FCS can still contain EVs (Lehrich *et al.* 2018), potentially affecting functional assays. Prior to our proliferation/suppression experiments using EVs, we had tested different cell culture media compositions for AMPase activity, and found that serum-supplemented media exhibited AMPase activity, while the serum-free X-VIVO 15 (optimized for T cell culture) did not (Response letter Fig. 1). Therefore, all our functional assays were performed in X-VIVO 15 medium (see material and methods section).

Response letter Fig. 1 | AMPase activity of FCS-supplemented cell culture medium and X-VIVO 15 T cell medium. Different cell culture media were incubated with 1,*N*⁶-etheno-AMP (eAMP) and generation of 1,*N*⁶-etheno-adenosine (eADO) was determined by HPLC, as described in the methods section of the manuscript. HS: human serum; FCS: fetal calf serum.

In order to determine the extent of protein contamination in our EV preparations and the influence of potentially existing medium-derived particles on our assays, we performed a mock isolation of EVs from X-VIVO 15 medium (referred to as X-VIVO UC pellet), following the same exact protocol as we did for our cell culture supernatants. Nanoparticle tracking analysis (NTA) did not detect any particles in the X-VIVO UC pellet. Still, to exclude any possible effect of contaminants (e.g. protein aggregates) and small particles not detectable by NTA, we performed a T cell proliferation assay adding the pellet from the X-VIVO 15 medium ultracentrifugation (new Supplementary Fig. 8a).

New Supplementary Fig. 8a | X-VIVO UC pellet does not affect T cell activation and proliferation. CD73⁻CD4con T cells (100,000 cells) were stimulated with α CD3/ α CD28 in the presence of the ADA inhibitor EHNA (10 μ M) and incubated with AMP (50 μ M), and rec. CD73 (15 ng/mL) or pelleted material after ultracentrifugation of X-VIVO 15 medium (X-VIVO UC pellet). CD25 expression and proliferation were measured after four days by flow cytometry (mean \pm SD of technical duplicates). UC: ultracentrifugation.

Addition of the X-VIVO UC pellet did not inhibit T cell activation or proliferation, and these results have been now included in the manuscript as new Supplementary Fig. 8a. In Fig. 5 of our manuscript we show that EVs from CD73⁻ CD8 T cells have no effect on T cell activation and proliferation, which also excludes any effect deriving from the cell culture medium. Moreover, the CD73-specific inhibitor PSB-14685 completely abolishes the suppressive effect of EVs derived from CD73⁺ CD8 T cells or from SF, indicating that the observed effect is specifically due to CD73 (Fig. 5 and Fig. 7e of the manuscript, respectively). Therefore, we conclude that contaminants in our EV samples do not contribute to the CD73-mediated T cell suppression described in our manuscript.

Protein contamination in SF-derived EVs isolated by ultracentrifugation has been reported (Foers *et al.* 2018). The major contaminant is albumin, which constitutes approximately 70% of the protein content in the SF (Guenther *et al.* 2014), and that is also abundant in serum-free media. The removal of contaminants from the EV preparation is still challenging and EV isolation methods are constantly being improved to combine highest purity with a high yield. Density gradient ultracentrifugation reduces the amount of contaminating albumin in the EV pellet; however, in our experimental setup we had to settle for a high yield and decided to use standard differential ultracentrifugation.

We assessed for the presence of albumin and apolipoproteins in EVs purified from T cell culture supernatants and SF by western blot, and found a small fraction of albumin in the pellets after high speed centrifugation of the X-VIVO 15 medium and of the SF. In contrast, apolipoproteins present in the medium and also in the SF, were eliminated after ultracentrifugation (new Supplementary Fig. 8b).

New Supplementary Fig. 8b | Detection of albumin and apolipoproteins in EV preparations by western blot.

Western blot analysis of albumin and lipoproteins in EVs derived from T cell culture medium and SF of JIA patients. Samples loaded: 3.5 µg per EV sample, 15 µl of X-VIVO UC pellet corresponding to 2 ml X-VIVO 15 medium, 15 µl X-VIVO 15 medium, 2 µl pure SF.

The figure shows that pelleted material from X-VIVO 15 medium did not suppress T cell activation and proliferation, even though albumin was detected in the EV preparations. This topic has now been introduced to the manuscript as the new Supplementary Fig. 8, and we have adapted the results section on page 10, lines 252-255:

‘Importantly, EVs isolated from the supernatant of CD73⁻ CD8 T cells did not suppress activation and proliferation of the responder T cells, and neither did the pelleted material after ultracentrifugation of cell culture medium (Fig. 5 and Supplementary Fig. 8), indicating that the suppressive effect observed is CD73-specific and not due to contaminants in the EV preparation.’

and on pages 11-12, lines 301-304:

‘We next isolated EVs from the SF and verified their EV nature by electron microscopy and western blot. The EV markers flotillin and CD81 could be detected in all EV samples along with CD73 (Supplementary Fig. 9a-b). We also assessed for the presence of protein contamination and found rests of albumin, as previously reported when using differential ultracentrifugation for EV isolation (Foers et al. 2018), but not of apolipoproteins (Supplementary Fig. 8b).’

Based on i) the absence of AMPase activity in X-VIVO 15 medium, ii) the lack of effect of X-VIVO UC pellets on T cell activation, iii) the fact that there is no influence of CD73⁻ EVs on T cell activation, and iv) the reversibility of the effect of CD73-containing EVs by the specific inhibitor PSB-14685, we conclude that the effects of EVs on T cell activation and proliferation are CD73-mediated.

2) On page 221, can the authors change the EV definition to ‘EV enriched pellet’ for the reason above.

As suggested, we have changed the EV definition on page 9, lines 215-216 to ‘EV-enriched pellet’ and also adapted the methods section accordingly (page 18, line 492) to make the reader aware of the protein contamination in the EV preparations.

3) *In the rebuttal the authors the authors mention go pg 4, 'they speculate that CD39-containing Treg-derived vesicles fuse with the membrane of the CD73-expressing effector cells, thus cooperating in the production of adenosine' and that 'Regarding the source of CD39, Tung et al. show that it is contained in Treg-derived EVs. We also show that the major ATPase activity (CD39) comes from Treg cells (Fig. 6a), and that Tregs, even in very small ratios to T effector cells, mediate immune suppression in the presence of CD73+ (but not CD73—) EVs (Fig. 6b)'. However this has not really been discussed in the discussion. I think that they have to modify the following statement in their discussion, 'and propose a coordinated effort involving CD39 on Tregs for the degradation of ATP to AMP, and CD73 on T cell-derived EVs that provide the necessary AMPase activity to generate adenosine', to include the fact that the Treg EVs have CD39 also and maybe contributing.*

We thank the reviewer for this suggestion. Because we did not perform experiments with Treg-derived EVs in the figures shown in the manuscript, we did not change the summary sentence at the beginning of the discussion. However, we agree with the reviewer that the expression and function of CD39 on Treg-derived EVs is highly interesting and relevant for our findings, and therefore we added the following paragraph to the discussion (page 15, lines 400-404).

'We have not addressed here the contribution of other ectonucleotidases present in T cell-derived EVs to ATP metabolism, but it has been shown that Treg-derived EVs contain CD39 and inhibit T cell proliferation (Tung et al. 2020). Further work will be required to dissect the role of cellular and EV-associated ectonucleotidases in immune suppression.'

4) *On pg 12 of the discussion, one reference Deaglio 2007 is not in the appropriate format.*

The reference "Deaglio, 2007" on page 12, line 324 has been corrected to the right format.

Response to Reviewer #2

It appears the authors have responded comprehensively to both reviewers. I would agree that some questions are beyond the scope of this manuscript.

We thank the reviewer for the positive feedback.

References

- Foers, Andrew D., Simon Chatfield, Laura F. Dagley, Benjamin J. Scicluna, Andrew I. Webb, Lesley Cheng, Andrew F. Hill, Ian P. Wicks, and Ken C. Pang. 2018. "Enrichment of Extracellular Vesicles from Human Synovial Fluid Using Size Exclusion Chromatography." *Journal of Extracellular Vesicles* 7 (1): 1490145. <https://doi.org/10.1080/20013078.2018.1490145>.
- Guenther, Leah E., Bryan W. Pyle, Thomas R. Turgeon, Eric R. Bohm, Urs P Wyss, Tannin A. Schmidt, and Jan-M Brandt. 2014. "Biochemical Analyses of Human Osteoarthritic and Periprosthetic Synovial Fluid." *Proceedings of the Institution of Mechanical Engineers, Part H: Journal of Engineering in Medicine* 228 (2): 127–39. <https://doi.org/10.1177/0954411913517880>.
- Lehrich, Brandon M., Yaxuan Liang, Pooya Khosravi, Howard J. Federoff, and Massimo S. Fiandaca. 2018. "Fetal Bovine Serum-Derived Extracellular Vesicles Persist within Vesicle-Depleted Culture Media." *International Journal of Molecular Sciences* 19 (11): 3538. <https://doi.org/10.3390/ijms19113538>.
- Tung, Sim Lai, Giorgia Fanelli, Robert Ian Matthews, Jordan Bazoer, Marilena Letizia, Gema Vizcay-Barrena, Farid N. Faruqu, et al. 2020. "Regulatory T Cell Extracellular Vesicles Modify T-Effector Cell Cytokine Production and Protect Against Human Skin Allograft Damage." *Frontiers in Cell and Developmental Biology* 8: 317. <https://doi.org/10.3389/fcell.2020.00317>.

REVIEWER COMMENTS

Reviewer #1 (Remarks to the Author):

Dear Authors, thank you for addressing all my concerns and points that I have raised. The changes you have made have strengthened your findings.